# OlympicArena: Benchmarking Multi-discipline Cognitive Reasoning for Superintelligent AI

**Zhen Huang**[3,4], **Zengzhi Wang**[1,4], **Shijie Xia**[1,4], **Xuefeng Li**[1,4], **Haoyang Zou**[4],
**Ruijie Xu**[1,4], **Run-Ze Fan**[1,4], **Lyumanshan Ye**[1,4], **Ethan Chern**[1,4], **Yixin Ye**[1,4], **Yikai Zhang**[1,4]
**Yuqing Yang**[4], **Ting Wu**[4], **Binjie Wang**[4], **Shichao Sun**[4], **Yang Xiao**[4], **Yiyuan Li**[4], **Fan Zhou**[1,4]
**Steffi Chern**[4], **Yiwei Qin**[4], **Yan Ma**[4], **Jiadi Su**[4], **Yixiu Liu**[1,4], **Yuxiang Zheng**[1,4]
**Shaoting Zhang**[2]*, **Dahua Lin**[2]*, **Yu Qiao**[2]*, **Pengfei Liu**[1,2,4]*

[1]Shanghai Jiao Tong University, [2]Shanghai Artificial Intelligence Laboratory,
[3]Soochow University, [4]Generative AI Research Lab (GAIR)

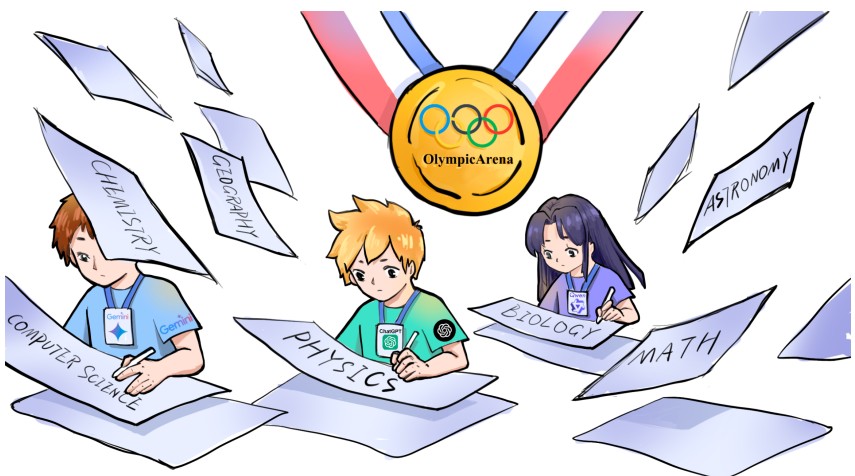

Figure 1: AI participates in the Olympics from the Gaokao [57] venue.

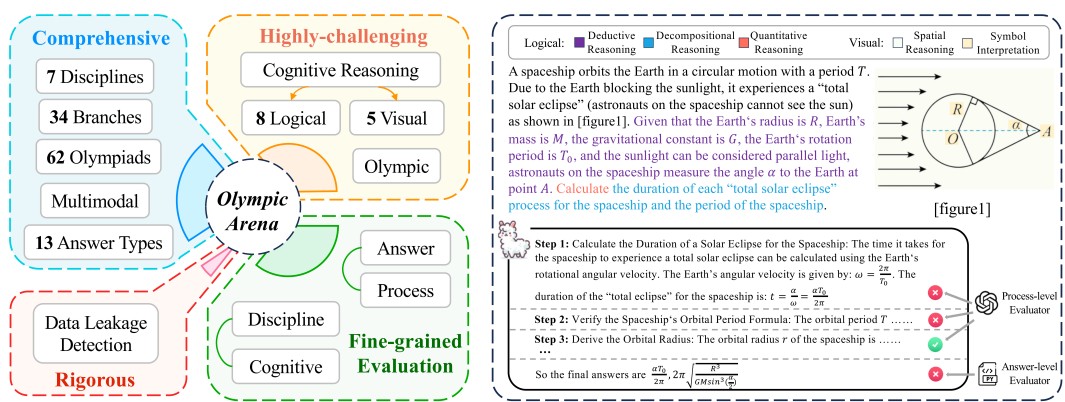

Figure 2: The overview of our *OlympicArena* benchmark.

*Corresponding authors

38th Conference on Neural Information Processing Systems (NeurIPS 2024) Track on Datasets and Benchmarks.

## Abstract

The evolution of Artificial Intelligence (AI) has been significantly accelerated by advancements in Large Language Models (LLMs) and Large Multimodal Models (LMMs), gradually showcasing potential *cognitive reasoning* abilities in problem-solving and scientific discovery (i.e., AI4Science) once exclusive to human intellect. To comprehensively evaluate current models' performance in cognitive reasoning abilities, we introduce *OlympicArena*, which includes 11,163 bilingual problems across both text-only and interleaved text-image modalities. These challenges encompass a wide range of disciplines spanning seven fields and 62 international Olympic competitions, rigorously examined for data leakage. We argue that the challenges in Olympic competition problems are ideal for evaluating AI's cognitive reasoning due to their complexity and interdisciplinary nature, which are essential for tackling complex scientific challenges and facilitating discoveries. Beyond evaluating performance across various disciplines using answer-only criteria, we conduct detailed experiments and analyses from multiple perspectives. We delve into the models' cognitive reasoning abilities, their performance across different modalities, and their outcomes in *process-level* evaluations, which are vital for tasks requiring complex reasoning with lengthy solutions. Our extensive evaluations reveal that even advanced models like GPT-4o only achieve a 39.97% overall accuracy (28.67% for mathematics and 29.71% for physics), illustrating current AI limitations in complex reasoning and multimodal integration. Through the *OlympicArena*, we aim to advance AI towards superintelligence, equipping it to address more complex challenges in science and beyond. We also provide a comprehensive set of resources to support AI research, including a benchmark dataset, an open-source annotation platform, a detailed evaluation tool, and a leaderboard with automatic submission features.[2]

## 1 Introduction

The landscape of Artificial Intelligence (AI) has undergone a transformative evolution with advances in technologies like Large Language Models [2, 3] and Large Multimodal Models (LMMs) [31]. These models represent significant milestones on the path to Artificial General Intelligence (AGI) [47, 15], demonstrating remarkable *cognitive reasoning* abilities, which represent drawing meaningful conclusions from incomplete and inconsistent knowledge to solve problems in complex scenarios [16, 34]. They adeptly handle tasks ranging from simple grade school math problems [13, 56, 59, 64] to complex challenges like those presented at the International Mathematical Olympiad (IMO) [46, 42]. Furthermore, they are progressively being applied to intricate real-world scenarios, such as using AI agents for software development [37], collaborating on complex decision-making processes [11] and even boosting the field of scientific research (i.e., AI4Science) [50].

These applications highlight AI's growing proficiency in cognitive reasoning, a crucial element in the pursuit of AGI and, potentially, superintelligence [35]. Therefore, how to benchmark these abilities has sparked extensive research. Existing benchmarks [18, 22, 26, 63, 44, 62] utilize multidisciplinary exam problems to assess the problem-solving skills of LLMs, but these problems are predominantly knowledge-intensive which has become relatively easy for current LLMs. Also, these benchmarks primarily focus on text-only modalities. Although some benchmarks begin to target college-level problems [52, 40] and incorporate multimodal assessments [58, 60, 61], they still predominantly focus on knowledge-intensive tasks or simple concept applications (shown in Table 1). Concurrent to our work, He et al. [17] introduces an Olympic-level benchmark yet it is limited to only mathematics and physics. Furthermore, all the above benchmarks lack a systematic and fine-grained evaluation of various cognitive reasoning abilities. For example, they mostly do the evaluation only based on answers, neglecting potential errors in the reasoning process. This underscores the need for more comprehensive evaluations that not only cover a broader range of disciplines but also focus on higher levels of cognitive reasoning as well as fine-grained evaluation.

In this paper, we introduce *OlympicArena*, a comprehensive, highly-challenging, and rigorously curated benchmark featuring a detailed, fine-grained evaluation mechanism designed to assess

---

[2]`https://github.com/GAIR-NLP/OlympicArena`

advanced AI capabilities across a broad spectrum of Olympic-level challenges (as illustrated in Figure 2). We extensively select, collect, and process problems from seven disciplines—mathematics, physics, chemistry, biology, geography, astronomy, and computer science—encompassing 62 different Olympic-level competitions. This extensive collection has culminated in a benchmark comprising 11,163 problems, categorized into 13 types of answers (e.g., expression, interval). Importantly, *OlympicArena* enhances its evaluation framework by incorporating *process-level evaluations* that scrutinize the step-by-step reasoning processes of AI models. This approach is critical for understanding the depth of cognitive reasoning beyond correct answers [29, 53], allowing us to identify and rectify gaps in AI reasoning pathways and ensuring more robust AI capabilities. The benchmark is bilingual, featuring both English and Chinese, to enhance its accessibility and global applicability. Additionally, it supports two modalities: text-only and interleaved text and images, catering to the evolving complexity of tasks that modern AI systems must handle. We also perform data leakage detection experiments [54] on some mainstream models to validate our benchmark's effectiveness.

We conduct a series of experiments across existing top-performing LMMs, encompassing both proprietary models (e.g., GPT-4o [36]) and open-source models (e.g., LLaVa-NeXT [31]). Additionally, we evaluate various types of LLMs (e.g., GPT-3.5) in two settings: text-only and image-caption and conduct comprehensive evaluations from both the answer-level and process-level perspectives. For answer-level evaluations, we combine rule-based and model-based (GPT-4V[3] in this paper) methods to cover a more diverse range of answer types. For process-level evaluations, we score each reasoning step of the model output, which we consider quite critical in reasoning scenarios. Additionally, we perform fine-grained evaluations and analyses on different types of cognitive reasoning, from both logical and visual perspectives to better interpret the current capabilities of AI.

Our observations from the *OlympicArena* benchmark are summarized as follows: (1) Even the most advanced model, GPT-4o, achieves only a 39.97% overall accuracy, while other open-source models struggle to reach a 20% overall accuracy, underscoring current models' limitations in handling complex, multidisciplinary problems that require advanced cognitive reasoning—key aspects of scientific discovery. (2) Through more fine-grained analysis § 4.4, we find that LMMs are particularly weak in handling complex, decompositional reasoning problems and exhibit poor spatial and geometric perception visual abilities, as well as difficulties in understanding abstract symbols. (3) Additionally, we discover that current LMMs seem to struggle significantly in leveraging interleaved visual information for complex cognitive reasoning problems. Various LMMs fail to show notable enhancements compared to their text-only counterparts. (4) The process-level evaluation also indicates that most models can correctly execute some reasoning steps in spite of providing incorrect answers, demonstrating the models' significant potential. (5) Through data leakage detection, we find that instances of data leakage in our benchmark are exceedingly rare. Even on the infrequent occasions when leakage does occur, the corresponding models do not consistently solve these problems correctly. This suggests the need for more advanced training strategies to enhance cognitive reasoning capabilities. These observations highlight the immense value of the *OlympicArena* benchmark in advancing our understanding of AI's capabilities and limitations.

## 2    Related Work

**Benchmark AI Intelligence**    How to benchmark AI intelligence has always been a challenging problem. Initially, the Turing Test [47] provided a conceptual framework for evaluating AI Intelligence. However, limitations in past AI technology lead researchers to focus on specialized domains. In computer vision, benchmarks like MNIST [25] and ImageNet [14] catalyze progress, while in natural language processing, GLUE [49] and XTREME [21] set the standard for evaluating linguistic capabilities across tasks and languages. The success of pretrained language models [38, 23] particularly recent LLMs emphasizes the evaluation of foundational knowledge and innate abilities as shown in Figure 2. This leads to the creation of benchmarks such as MMLU [18], AGIEval[63], C-Eval [22], and CMMLU [26], which pushed the limits of language models with multidisciplinary, multilingual, and knowledge-intensive tasks. However, the rapid progress of LLMs has rendered these benchmarks insufficient to fully assess the models' growing capabilities.

---

[3]At the time of doing most part of this work, GPT-4o has not been released yet, so GPT-4V is mainly used for annotating, evaluation, and case study.

Table 1: Comparison of various benchmarks. "Subjects": ■ Math, ■ Physics, ■ Chemistry, ■ Biology, ■ Geography, ■ Astronomy, ■ Computer Science. "Multimodal" indicates whether the benchmark contains visual information. "Language": "EN" for English and "ZH" for Chinese. "Size" represents the number of test problems. "#Answer" shows the number of answer types. "Eval." details evaluation methods: ■ rule-based, ■ model-based, ■ answer-level, ■ process-level. "Leak Det." indicates if data leakage detection is conducted. "Difficulty" shows problem proportions at difficulty levels: ■ Knowledge Recall, ■ Concept Application, ■ Cognitive Reasoning. "#Logic." indicates the average logical reasoning abilities per question, and "#Visual." indicates the average visual reasoning abilities per multimodal question.

| Benchmark | Subjects | Multimodal | Language | Size | #Answer | Eval. | Leak Det. | Difficulty | #Logic. | #Visual. |
|---|---|---|---|---|---|---|---|---|---|---|
| SciBench | | ✓ | EN | 789 | 1 | ■ / ■ | × | | 0.39 | 2.35 |
| CMMLU | | × | ZH | 1594 | 1 | ■ / ■ | × | | 0.36 | - |
| MMLU | | × | EN | 2554 | 1 | ■ / ■ | × | | 0.44 | - |
| C-Eval | | × | ZH | 3362 | 1 | ■ / ■ | × | | 0.6 | - |
| MMMU | | ✓ | EN | 3007 | 2 | ■ / ■ | × | | 0.25 | 2.75 |
| SciEval | | × | EN | 15901 | 4 | ■ / ■ | × | | 1.12 | - |
| AGIEval | | × | EN & ZH | 3300 | 2 | ■ / ■ | × | | 1.07 | - |
| GPQA | | × | EN | 448 | 1 | ■ / ■ | × | | 2.24 | - |
| JEEBench | | × | EN | 515 | 3 | ■ / ■ | × | | 2.41 | - |
| OlympiadBench | | ✓ | EN & ZH | 8952 | 7 | ■ / ■ | × | | 2.26 | 2.96 |
| **OlympicArena** | | ✓ | EN & ZH | 11163 | **13** | ■ ■ / ■ ■ | ✓ | | **2.73** | **3.15** |

**Cognitive Reasoning** is crucial as it allows AI systems to apply prior knowledge and logical principles to complex tasks in a more human-like manner, ensuring better robustness and generalization in real-world applications [43]. Thus, more attention is paid to more intricate reasoning tasks, benchmarks like GSM8K [13] focused on grade-school mathematical reasoning problems, while MATH [20] introduced high-school level mathematical competition tasks. Furthermore, benchmarks such as JEEBench [4], SciBench [52], GPQA [40] and MMMU [58] have expanded the scope by incorporating multidisciplinary university-level subjects and even multimodal tasks. To further challenge AI systems, researchers have turned to problems from some of the most difficult competitions, specifically International Olympiads [17, 46, 30] and algorithmic challenges [28, 19, 41]. Nevertheless, there is currently no Olympic-level, multidisciplinary benchmark that comprehensively evaluates comprehensive problem-solving abilities to fully test all-rounded AI's cognitive ability. Table 1 presents a comparison of several related scientific benchmarks.

**Rigorous Evaluation for Reasoning** While curating comprehensive and appropriate data is crucial in benchmarks, adopting rigorous evaluation methodologies is equally important. Most existing benchmarks, as mentioned above, primarily focus on answer-level evaluation (i.e., only comparing the model's output with the standard answer). Recently, some works have started to focus on the models' intermediate reasoning steps. Some of them [48, 29, 51] explore using process supervision to train better reward models. Lanham et al. [24] delves into the faithfulness of the chain-of-thought reasoning process, while Xia et al. [53] trains models specifically designed to evaluate the validity and redundancy of reasoning steps for mathematical problems. However, in the evaluation methodologies of existing benchmarks as listed in Table 1, few of them incorporate process-level evaluation. This insufficient evaluation often neglects the reliability and faithfulness of AI models, especially in complex cognitive reasoning scenarios requiring lengthy solutions. In this work, the introduced *OlympicArena* is equipped with a more fine-grained evaluation methodology (i.e., process-level evaluation), allowing developers to better understand the true reasoning behaviors of models.

## 3 The OlympicArena Benchmark

### 3.1 Overview

We introduce the *OlympicArena*, an Olympic-level, multidisciplinary benchmark designed to rigorously assess the cognitive reasoning abilities of LLMs and LMMs. Our benchmark features a combination of text-only and interleaved text-image modalities, presented bilingually to promote accessibility and inclusivity. It spans seven core disciplines: **mathematics**, **physics**, **chemistry**, **biology**, **geography**, **astronomy**, and **computer science**, encompassing a total of 34 specialized branches (details are in Appendix A.1) which represent fundamental scientific fields.

The benchmark includes a comprehensive set of 11,163 problems from 62 distinct Olympic competitions, structured with 13 answer types (shown in Appendix A.2) from objective types (e.g., multiple choice and fill-in-the-blanks) to subjective types (e.g., short answers and programming tasks), which distinguishes it from many other benchmarks that primarily focus on objective problems. Detailed statistics of *OlympicArena* are described in Table 2. Also, to identify potential data leakage, we conduct specialized data leakage detection experiments on several models.

Furthermore, in pursuit of a granular analysis of model performance, we categorize cognitive reasoning into 8 types of logical reasoning abilities and 5 types of visual reasoning abilities. This comprehensive categorization aids in the detailed evaluation of the diverse and complex reasoning skills that both LLMs and LMMs can exhibit. Additionally, we specifically investigate all multimodal problems to compare the performance of LMMs against their text-based counterparts, aiming to better assess LMMs' capabilities in handling visual information. Finally, we evaluate the correctness and efficiency of the reasoning process, not just limited to an answer-based assessment.

Table 2: Benchmark Statistics

| Statistic | Number |
| --- | --- |
| Total Problems | 11163 |
| Total Competitions | 62 |
| Total Subjects/Subfields | 7/34 |
| Total Answer Types | 13 |
| Problems with Solutions | 7904 |
| Language (EN: ZH) | 7054: 4109 |
| Total Images | 7571 |
| Problems with Images | 4960 |
| Image Types | 5 |
| Cognitive Complexity Levels | 3 |
| Logical Reasoning Abilities | 8 |
| Visual Reasoning Abilities | 5 |
| Average Problem Tokens | 244.8 |
| Average Solution Tokens | 417.1 |

## 3.2 Data Collection

To ensure comprehensive coverage of Olympic-level problems across various disciplines, we begin by collecting URLs of various competitions where problems are publicly available for download in PDF format. Then, we utilize the Mathpix[4] tool to convert these PDF documents into markdown format, making them compatible with input requirements for models. Specifically, for the programming problems of Computer Science, we additionally collect corresponding test cases. We strictly adhere to copyright and licensing considerations, ensuring compliance with all relevant regulations.

## 3.3 Data Annotation

**Problem Extraction and Annotation.** To extract individual problems from the markdown format of the test papers, we employ about 30 students with background in science and engineering. We have developed a user interface for annotating multimodality data, which has been released. [5] To facilitate further research and the process-level evaluation of models, we annotate meta-information like solutions if provided. To ensure data quality, we implement a multi-step validation process after the initial annotation is completed. More details can be seen in Appendix B.1. After collecting all the problems, we perform deduplication within each competition based on model embeddings to remove repeated problems that may appear in multiple test papers from the same year. To further demonstrate that our benchmark emphasizes cognitive reasoning more than most other benchmarks, we categorize the difficulty of the problems into three levels and make comparison with other related benchmarks. Specifically, we classify all problems into: *knowledge recall*, *concept application* and *cognitive reasoning*. We utilize GPT-4V as the annotator for categorizing different difficulty levels[6] (detailed definitions and specific prompts can be found in Appendix B.2).[7]

**Annotation of Cognitive Reasoning Abilities.** To facilitate better fine-grained analysis, we categorize cognitive reasoning abilities from both logical and visual perspectives [16, 43]. The logical reasoning abilities encompass *Deductive Reasoning (DED)*, *Inductive Reasoning (IND)*, *Abductive Reasoning (ABD)*, *Analogical Reasoning (ANA)*, *Cause-and-Effect Reasoning (CAE)*, *Critical Thinking (CT)*, *Decompositional Reasoning (DEC)*, and *Quantitative Reasoning (QUA)*. Meanwhile, the visual reasoning abilities include *Pattern Recognition (PR)*, *Spatial Reasoning (SPA)*, *Diagrammatic Reasoning (DIA)*, *Symbol Interpretation (SYB)*, and *Comparative Visualization (COM)*. We also utilize GPT-4V as the annotator for categorizing different cognitive abilities (detailed definitions and

---

[4] https://mathpix.com/

[5] https://github.com/GAIR-NLP/OlympicArena/tree/main/annotation

[6] We annotate the validation sets to highlight their characteristics and save costs.

[7] All annotations using GPT-4V are manually verified for reliability.

specific prompts can be found in Appendix B.3).[7] With these annotations, we can conduct a more fine-grained analysis of the current cognitive reasoning abilities of AI.

### 3.4  Data Splitting

Our benchmark includes 11,163 problems, with 548 designated for model-based evaluation as *OlympicArena-ot*. We sample 638 problems across subjects to create *OlympicArena-val* for hyper-parameter tuning or small-scale testing. *OlympicArena-val* problems have step-by-step solutions, supporting research like process-level evaluation. The remaining problems form *OlympicArena-test*, the official test set with unreleased answers for formal testing. The results in this paper are based on the entire benchmark dataset, including *OlympicArena-ot*, *OlympicArena-val*, and *OlympicArena-test*.

## 4  Experiments

### 4.1  Experimental Setup

To comprehensively evaluate the capabilities of LLMs and LMMs (selected models are listed in Appendix C.2) across different modalities, we design our experiments to include three distinct settings: multimodal, image-caption, and text-only. In the multimodal setting, we assess the ability of LMMs to leverage visual information by interleaving text and images, simulating real-world scenarios. For models unable to handle interleaved inputs, we concatenate multiple images into a single input. For LMMs requiring necessary image inputs, their text-based counterparts handle text-only problems. In the image-caption setting, we explore whether textual descriptions of images enhance the problem-solving capabilities of LLMs. Using InternVL-Chat-V1.5[8] [12], we generate captions for all images based on prompts detailed in Appendix C.1. These captions replace the original image inputs. In the text-only setting, we evaluate the performance of LLMs without any visual information, serving as a baseline to compare against the multimodal and image-caption settings. All experiments use zero-shot prompts, tailored to each answer type and specifying output formats to facilitate answer extraction and rule-based matching. It also minimizes biases typically associated with few-shot learning [32, 33]. Detailed prompt designs are provided in Appendix C.3.

### 4.2  Evaluation

**Answer-level Evaluation**  We combine rule-based and model-based methods to cover a diverse range of problems. For problems with fixed answers, we extract the final answer and perform rule-based matching according to the answer type. For code generation tasks, we use the unbiased pass@k metric [10] to test all test cases. For problems with answer types categorized as "others" which are difficult to be evaluated using rule-based matching (e.g., chemical equation writing problems), we employ GPT-4V as an evaluator to assess the responses. To ensure the reliability of GPT-4V as an evaluator, we manually sample and check the correctness. More details will be explained in Appendix C.5.

**Process-level Evaluation**  To further investigate the correctness of the reasoning steps, ensuring a rigorous assessment of the cognitive abilities of models, we conduct the process-level evaluation. We first sample 96 problems with reference solutions from *OlympicArena*. We employ GPT-4 to convert both the references (i.e., gold solutions) and the model-generated solutions into a structured step-by-step format. We then provide these solutions to GPT-4V and score each step for its correctness on a scale ranging from 0 to 1. [9] The experimental details can be seen in Appendix C.6. To validate the consistency with human judgment, we obtain some samples for human annotations. The results indicate that our model-based evaluation method is highly accurate, with an 83% inter-annotator agreement.

To further concretize models' performance, we sample a portion of the problems where the model makes errors and conduct an **error analysis** on them. Details can be found in Appendix D.5.

Table 3: Experimental results on *OlympicArena*, expressed as percentages, with the highest score in each setting underlined and the highest scores across all settings bolded. We use the pass@k metric (Equation 1) for CS problems. When calculating the overall accuracy, for code generation problems, if any generated code for a problem passes all test cases, the problem is considered correct.

| Model | Math | Physics | Chemistry | Biology | Geography | Astronomy | CS | Overall |
|---|---|---|---|---|---|---|---|---|
| | Accuracy | Accuracy | Accuracy | Accuracy | Accuracy | Accuracy | Pass@1 | Accuracy |
| LLMs | | | | | | | | |
| Qwen-7B-Chat | 1.58 | 3.74 | 7.01 | 7.31 | 4.53 | 5.48 | 0 | 4.31 |
| Yi-34B-Chat | 3.06 | 9.77 | 23.53 | 32.67 | 35.03 | 18.15 | 0.17 | 17.31 |
| Internlm2-20B-Chat | 5.88 | 9.48 | 18.36 | 31.90 | 32.14 | 16.03 | 0.60 | 16.62 |
| Qwen1.5-32B-Chat | 9.65 | 14.54 | 29.84 | 38.58 | 40.69 | 28.05 | 0.51 | 23.69 |
| GPT-3.5 | 7.27 | 10.92 | 23.03 | 31.19 | 31.13 | 16.93 | 3.85 | 18.27 |
| Claude3 Sonnet | 7.76 | 17.24 | 29.46 | 38.25 | 40.94 | 24.04 | 1.62 | 23.02 |
| GPT-4 | 19.46 | 24.77 | 42.52 | 46.47 | 44.97 | 33.44 | 7.78 | 32.37 |
| GPT-4o | 28.33 | 29.54 | 46.24 | 49.42 | 48.36 | 43.25 | 8.46 | 38.17 |
| Image caption + LLMs | | | | | | | | |
| Qwen-7B-Chat | 1.76 | 3.56 | 6.75 | 7.83 | 7.17 | 6.87 | 0 | 4.89 |
| Yi-34B-Chat | 3.01 | 9.94 | 21.45 | 31.26 | 34.78 | 17.33 | 0.17 | 16.72 |
| Internlm2-20B-Chat | 5.94 | 10.40 | 20.25 | 31.00 | 32.52 | 16.93 | 0.73 | 17.07 |
| Qwen1.5-32B-Chat | 9.56 | 14.31 | 29.84 | 38.51 | 40.75 | 27.2 | 0.60 | 23.43 |
| GPT-3.5 | 7.16 | 14.48 | 23.97 | 30.94 | 33.52 | 18.56 | 4.70 | 18.83 |
| Claude3 Sonnet | 7.52 | 18.10 | 29.84 | 38.77 | 41.14 | 22.65 | 2.39 | 23.10 |
| GPT-4 | 19.46 | 26.21 | 41.58 | 45.89 | 48.18 | 35 | 7.63 | 33.00 |
| GPT-4o | 28.27 | 29.71 | 45.87 | 51.16 | 49.12 | 43.17 | **9.57** | 38.50 |
| LMMs | | | | | | | | |
| Qwen-VL-Chat | 1.73 | 4.25 | 8.64 | 12.13 | 13.77 | 7.85 | 0 | 6.90 |
| Yi-VL-34B | 2.94 | 9.94 | 19.81 | 27.73 | 25.16 | 16.60 | 0 | 14.49 |
| InternVL-Chat-V1.5 | 6.03 | 9.25 | 19.12 | 30.39 | 32.96 | 15.94 | 0.38 | 16.63 |
| LLaVA-NeXT-34B | 3.03 | 10.06 | 21.45 | 33.18 | 36.92 | 18.15 | 0.18 | 17.38 |
| Qwen-VL-Max | 6.93 | 12.36 | 23.79 | 36 | 40.19 | 23.39 | 0.77 | 20.65 |
| Gemini Pro Vision | 6.28 | 12.47 | 28.14 | 37.48 | 37.42 | 20.20 | 1.45 | 20.97 |
| Claude3 Sonnet | 7.52 | 18.16 | 29.27 | 38.96 | 40.13 | 25.02 | 1.45 | 23.13 |
| GPT-4V | 19.27 | 24.83 | 41.45 | 46.79 | 49.62 | 32.46 | 7.00 | 32.76 |
| GPT-4o | **28.67** | **29.71** | **46.69** | **52.18** | **56.23** | **43.91** | 9.00 | **39.97** |

## 4.3 Main Results

Table 3 presents the evaluation results of various LMMs and LLMs on *OlympicArena*. We obtain the following observations: (1) Even the most advanced large model, GPT-4o, achieves only a 39.97% overall accuracy, while other open-source models struggle to reach a 20% overall accuracy. This stark contrast highlights the significant difficulty and rigor of our benchmark, demonstrating its effectiveness in pushing the boundaries of current AI capabilities. (2) Furthermore, compared to subjects like biology and geography, we observe that mathematics and physics remain the two most challenging disciplines, likely due to their reliance on complex reasoning abilities. (3) Computer programming competitions also prove to be highly difficult, with some open-source models failing to solve any of them, indicating current models' poor abilities to design efficient algorithms to solve complex problems.

## 4.4 Fine-grained Analysis

To achieve a more fine-grained analysis of the experimental results, we conduct further evaluations based on different modalities and reasoning abilities. Additionally, we also conduct an analysis of the process-level evaluation. Key findings are as follows:

**Models exhibit varied performance across different logical and visual reasoning abilities.** As shown in Figure 3, almost all models demonstrate similar performance trends across different logical reasoning abilities. They tend to excel in Abductive Reasoning and Cause-and-Effect Reasoning, doing well in identifying causal relationships from the provided information. Conversely, models perform poorly in Inductive Reasoning and Decompositional Reasoning. This is due to the diverse

---

[8]We use InternVL-Chat-V1.5 for its high performance and cost-effective captioning.

[9]We leave more research on open-source model-based evaluation for future work.

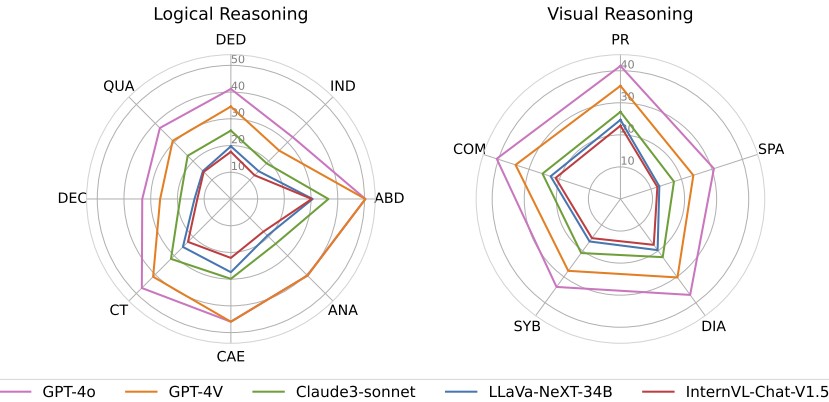

Figure 3: Performance of various models on logical and visual reasoning abilities. Logical reasoning abilities: Deductive Reasoning (DED), Inductive Reasoning (IND), Abductive Reasoning (ABD), Analogical Reasoning (ANA), Cause-and-Effect Reasoning (CAE), Critical Thinking (CT), Decompositional Reasoning (DEC), and Quantitative Reasoning (QUA). Visual reasoning abilities: Pattern Recognition (PR), Spatial Reasoning (SPA), Diagrammatic Reasoning (DIA), Symbol Interpretation (SYB), and Comparative Visualization (COM).

and unconventional nature of Olympic-level problems, which require the ability to break down complex problems into smaller sub-problems. In terms of visual reasoning abilities, models tend to be better at Pattern Recognition and Comparative Visualization. However, they struggle with tasks involving spatial and geometric reasoning as well as those need to understand abstract symbols. The completed results are presented in Appendix D.1.

**Most LMMs are still not proficient at utilizing visual information.** As displayed in Figure 4a, only a few LMMs (such as GPT-4o and Qwen-VL-Chat) show significant improvement with image inputs compared to their text-based counterpart. Many LMMs do not exhibit enhanced performance with image inputs and some even show decreased effectiveness when handling images. Possible reasons include: (1) When text and images are input together, LMMs may focus more on the text, neglecting the information in the images. This conclusion has also been found in some other works [61, 9]. (2) Some LMMs, while training their visual capabilities based on their text-based models, may lose some of their inherent language abilities (e.g., reasoning abilities), which is particularly evident in our scenarios. (3) Our problems use a complex interleaved text and image format, which some models do not support well, leading to difficulties in processing and understanding the positional information of images embedded within the text. [10]

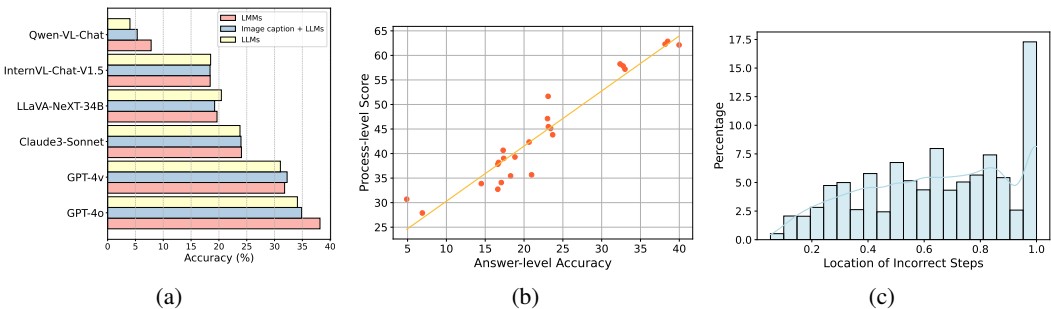

(a)  (b)  (c)

Figure 4: (a) Comparison of different LMMs and their corresponding LLMs across three different experimental settings. For details on the corresponding LLMs for each LMM, refer to the Appendix C.2. (b) The correlation between answer-level and process-level scores of all the models over all the sampled problems. (c) Distribution of the locations of incorrect steps, represented as the proportion of steps from left to right in the entire process, over all the sampled problems.

---

[10]We exclude Yi-VL-34B as it doesn't support multiple image inputs, which may cause an unfair comparison.

**Analysis of process-level evaluation results** Through process-level evaluation (complete results are in Table 14), we discover following insights: (1) There is generally a high consistency between process-level evaluation and answer-level evaluation. When a model produces a correct answer, the quality of the reasoning process tends to be higher most of the time (see Figure 4b). (2) The accuracy at the process-level is often higher than at the answer-level. This indicates that even for very complex problems, the model can correctly perform some of the intermediate steps. Therefore, the model likely has significant untapped potential for cognitive reasoning, which opens new avenues for researchers to explore. We also find that in a few disciplines, some models that perform well at the answer level fall behind at the process level. We speculate that this is because models sometimes tend to overlook the reasonableness of intermediate steps when generating answers, even though these steps may not be crucial to the final result. (3) Additionally, we conduct a statistical analysis of the location distribution of error steps (see Figure 4c). We identify that a higher proportion of errors occur in the later stages. This suggests that models are more prone to making mistakes as reasoning accumulates, indicating a need for improvement in handling long chains of logical deductions.

## 4.5  Efforts on Data Leakage Detection

Given the increasing scale of pre-training corpora, it is crucial to detect potential benchmark leakage. The opacity of pre-training often makes this task challenging. To this end, we employ a recently proposed instance-level leakage detection metric, *N-gram Prediction Accuracy* [54]. This metric uniformly samples several starting points for each instance, predicts the next n-gram for each starting point, and checks whether all predicted n-grams are correct, indicating that the model has potentially encountered this instance. We apply this metric to all available base or text-only chat models of the evaluated models. As shown in Figure 5, it is surprising yet reasonable that some base models or text-only chat models behind these evaluated models have potentially encountered a few benchmark instances, although the number is negligible compared to the complete benchmark. For

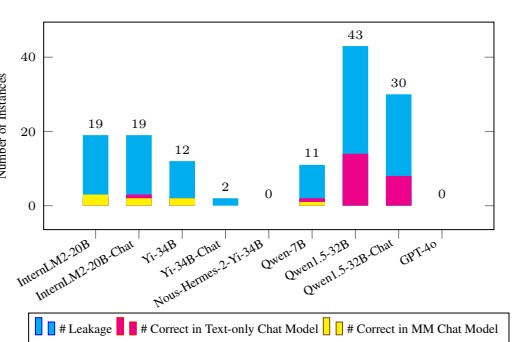

Figure 5: Detected number of leaked samples and the number of correct responses by corresponding text-only and multimodal chat models on these samples.

instance, the base model of Qwen1.5-32B-Chat has potentially encountered 43 benchmark instances. Furthermore, this raises a natural question: *can the model correctly answer these instances?* Interestingly, the corresponding text-only chat models and multimodal chat models can correctly answer even fewer of these instances. These results demonstrate that our benchmark has minimal leakage[11] and is sufficiently challenging, as the models cannot correctly answer most of the leaked instances. See Appendix E for more results and analysis.

## 5  Conclusion

In this work, we introduce *OlympicArena*, a comprehensive benchmark for evaluating the cognitive reasoning abilities of LMMs and LLMs on Olympic-level problems. Through our detailed experiments, we find that even the most powerful model at present, GPT-4o, does not perform well in applying cognitive reasoning abilities to solve complex problems. We hope that our OlympicArena benchmark serves as a valuable stepping stone for future advancements in AI for science and engineering.

---

[11]We also look forward to the development of more advanced detection tools and approaches.

## Acknowledgments and Disclosure of Funding

This work is partially supported by the National Natural Science Foundation of China (62476168) and the Shanghai Artificial Intelligence Laboratory, whose support has greatly contributed to the advancement of our research.

We sincerely appreciate all the laboratory members for their contributions in data annotation, project discussions, and providing valuable suggestions. Additionally, we extend our gratitude to Teacher Xiaoxia Yu from Hefei No. 168 Middle School for providing us with extensive information on various subjects. We also thank everyone who helps annotate the data for our benchmark dataset.

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

# A  Detailed Statistics of the Benchmark

## A.1  Distribution of Problems

Our benchmark collects data from various competitions. The detailed list can be found in Table 4. Note that a small portion of the problems are sampled from other related benchmarks which are marked in the table. The subfields covered by each competition subject are shown in Table 5. Additionally, the distribution information of our benchmark across different languages and modalities is presented in Table 6.

## A.2  Answer Types

Through extensive observation of a large number of problems and a thorough examination of multiple previous benchmarks, we have finally distilled 13 comprehensive answer types. These types are designed to cover as many problems as possible. The specific definitions for each answer type are provided in Table 7.

## A.3  Image Types

We categorize and summarize the five most common types of images in our multimodal scientific problems. The definitions of these types can be found in Table 8, and examples are provided in Figure 6. The distribution of different image types in our benchmark is shown in Figure 7

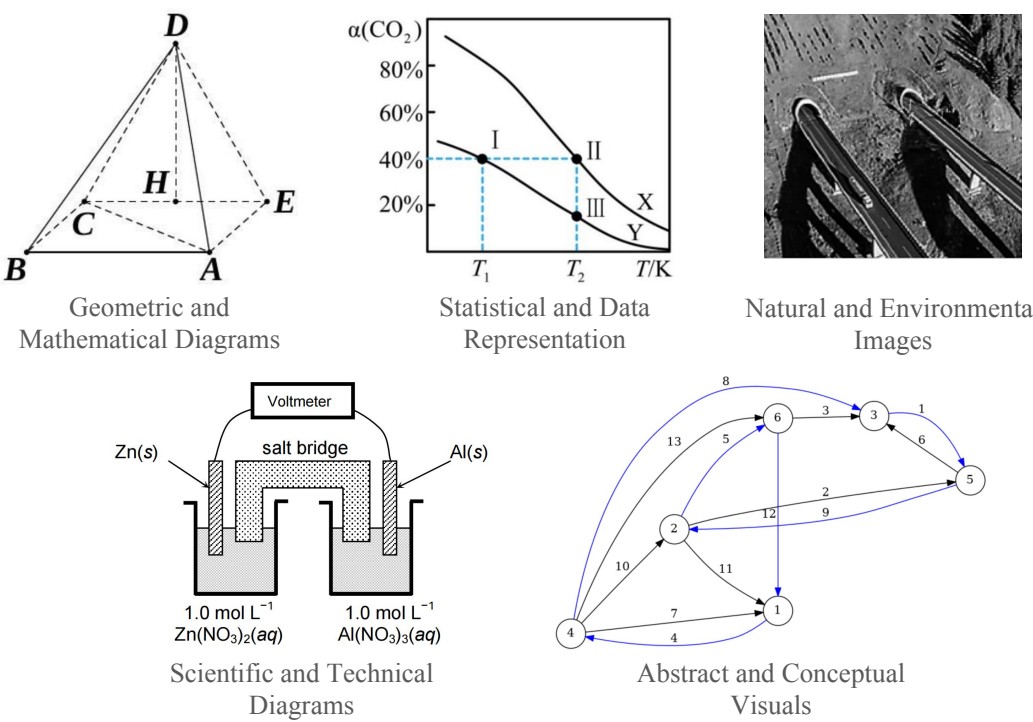

Figure 6: Examples of Image Types

Table 4: List of competitions included in OlympicArena. Competitions marked with * are partially sourced from OlympiadBench [17], and those marked with † are partially sourced from MMcode [27].

| Competition Name | Abbreviation | Subject | # Problems |
|---|---|---|---|
| UK Senior Kangaroo | UKMT_SK | Math | 20 |
| Math Majors of America Tournament for High Schools | MMATHS | Math | 47 |
| Math Kangaroo | MK | Math | 35 |
| Euclid Mathematics Contest | EMC | Math | 215 |
| Canadian Open Mathematics Challenge | COMC | Math | 26 |
| Johns Hopkins Mathematics Tournament | JHMT | Math | 100 |
| Berkeley Math Tournament | BMT | Math | 93 |
| Stanford Mathematics Tournament | SMT | Math | 473 |
| Chinese High School Mathematics League (Pre Round) | ZH_Math_PRE | Math | 546 |
| Chinese High School Mathematics League (1st&2nd Round) | ZH_Math_12 | Math | 279 |
| Duke University Math Meet | DMM | Math | 107 |
| The Princeton University Mathematics Competition | PUMaC | Math | 296 |
| Harvard-MIT Mathematics Tournament | HMMT | Math | 392 |
| William Lowell Putnam Mathematics Competition | Putnam | Math | 136 |
| International Mathematical Olympiad* | IMO | Math | 79 |
| Romanian Master of Mathematics* | RMM | Math | 8 |
| American Regions Mathematics League* | ARML | Math | 374 |
| Euclid Mathematics Competition* | EMC | Math | 215 |
| European Girls' Mathematical Olympiad* | EGMO | Math | 7 |
| F=MA | FMA | Physics | 122 |
| Intermediate Physics Challenge (Y11) | BPhO_IPC | Physics | 50 |
| Senior Physics Challenge | BPhO_SPC | Physics | 38 |
| Australian Science Olympaids Physics | ASOP | Physics | 48 |
| European Physics Olympiad | EPhO | Physics | 15 |
| Nordic-Baltic Physics Olympiad | NBPhO | Physics | 102 |
| World Physics Olympics | WoPhO | Physics | 38 |
| Asian Physics Olympiad | APhO | Physics | 126 |
| International Physics Olympiad | IPhO | Physics | 307 |
| Canadian Association of Physicists | CAP | Physics | 100 |
| Physics Bowl | PB | Physics | 100 |
| USA Physics Olympiad | USAPhO | Physics | 188 |
| Chinese Physics Olympiad | CPhO | Physics | 462 |
| Physics Challenge (Y13) | PCY13 | Physics | 44 |
| Chinese High School Biology Challenge | GAOKAO_Bio | Biology | 652 |
| International Biology Olympiad | IBO | Biology | 300 |
| The USA Biology Olympiad | USABO | Biology | 96 |
| Indian Biology Olympiad | INBO | Biology | 86 |
| Australian Science Olympiad Biology | ASOB | Biology | 119 |
| British Biology Olympiad | BBO | Biology | 82 |
| New Zealand Biology Olympiad | NZIBO | Biology | 223 |
| Chem 13 News | Chem13News | Chemistry | 56 |
| Avogadro | Avogadro | Chemistry | 55 |
| U.S. National Chemistry Olympiad (local) | USNCO (local) | Chemistry | 54 |
| U.S. National Chemistry Olympiad | USNCO | Chemistry | 98 |
| Chinese High School Chemistry Challenge | GAOKAO_Chem | Chemistry | 568 |
| Canadian Chemistry Olympic | CCO | Chemistry | 100 |
| Australian Science Olympiad Chemistry | ASOC | Chemistry | 91 |
| Cambridge Chemistry Challenge | C3H6 | Chemistry | 61 |
| UK Chemistry Olympiad | UKChO | Chemistry | 100 |
| International Chemistry Olympiad | IChO | Chemistry | 402 |
| Chinese High School Geography Challenge | GAOKAO_Geo | Geography | 862 |
| US Earth Science Organization | USESO | Geography | 301 |
| Australian Science Olympiad Earth Science | ASOE | Geography | 100 |
| The International Geography Olympaid | IGeO | Geography | 327 |
| Chinese High School Astronomy Challenge | GAOKAO_Astro | Astronomy | 740 |
| The International Astronomy and Astrophysics Competition | IAAC | Astronomy | 50 |
| USA Astronomy and Astrophysics Organization | USAAAO | Astronomy | 100 |
| British Astronomy and Astrophysics Olympaid Challenge | BAAO_challenge | Astronomy | 148 |
| British Astronomy and Astrophysics Olympaid-round2 | BAAO | Astronomy | 185 |
| USA Computing Olympiad | USACO | CS | 48 |
| Atcoder | Atcoder | CS | 48 |
| Codeforces† | CF | CS | 138 |

Table 5: Subfields of each subject included in OlympicArena.

| Subject | Subfields |
|---|---|
| Math | Algebra, Geometry, Number Theory, Combinatorics |
| Physics | Mechanics, Electricity and Magnetism, Waves and Optics, Thermodynamics, Modern Physics, Fluid Mechanics |
| Chemistry | General Chemistry, Organic Chemistry, Inorganic Chemistry, Analytical Chemistry, Physical Chemistry, Environmental Chemistry |
| Biology | Cell biology, Plant Anatomy and Physiology, Animal Anatomy and Physiology, Ethology, Genetics and Evolution, Ecology , Biosystematics |
| Geography | Physical Geography, Human Geography, Regional Geography, Environmental Geography, Geospatial Techniques |
| Astronomy | Fundamentals of Astronomy, Stellar Astronomy, Galactic and Extragalactic Astronomy, Astrophysics |
| CS | Data Structures, Algorithm |

Table 6: Statistics of OlympicArena benchmark across different disciplines and modalities.

|  | Mathematics | Physics | Chemistry | Biology | Geography | Astronomy | CS |
|---|---|---|---|---|---|---|---|
| EN & text | 2215 | 632 | 782 | 352 | 211 | 219 | 90 |
| EN & multi-modal | 193 | 646 | 235 | 554 | 517 | 264 | 144 |
| ZH & text | 780 | 164 | 124 | 312 | 58 | 264 | 0 |
| ZH & multi-modal | 45 | 298 | 444 | 340 | 804 | 476 | 0 |
| Total EN | 2408 | 1278 | 1017 | 906 | 728 | 483 | 234 |
| Total ZH | 825 | 462 | 568 | 652 | 862 | 740 | 0 |
| Total text | 2995 | 796 | 906 | 664 | 269 | 483 | 90 |
| Total multi-modal | 238 | 944 | 679 | 894 | 1321 | 740 | 144 |
| Grand Total | 3233 | 1740 | 1585 | 1558 | 1590 | 1223 | 234 |

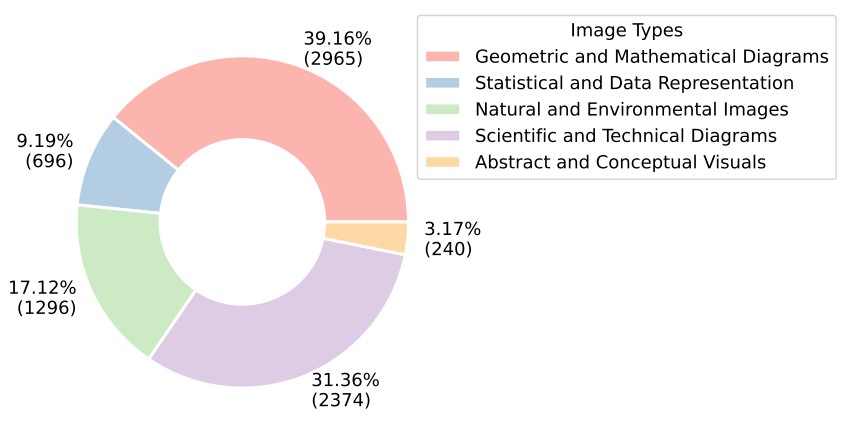

Figure 7: Distribution of Image Types

Table 7: Answer Types and Definitions

| Answer Type | Definition |
| --- | --- |
| Single Choice (SC) | Problems with only one correct option (e.g., one out of four, one out of five, etc.). |
| Multiple-choice (MC) | Problems with multiple correct options (e.g., two out of four, two out of five, two out of six, etc.). |
| True/False (TF) | Problems where the answer is either True or False. |
| Numerical Value (NV) | Problems where the answer is a numerical value, including special values like $\pi$, $e$, $\sqrt{7}$, $\log_2 9$, etc., represented in LaTeX. |
| Set (SET) | Problems where the answer is a set, such as $\{1, 2, 3\}$. |
| Interval (IN) | Problems where the answer is a range of values, represented as an interval in LaTeX. |
| Expression (EX) | Problems requiring an expression containing variables, represented in LaTeX. |
| Equation (EQ) | Problems requiring an equation containing variables, represented in LaTeX. |
| Tuple (TUP) | Problems requiring a tuple, usually representing a pair of numbers, such as (x, y). |
| Multi-part Value (MPV) | Problems requiring multiple quantities to be determined within a single sub-problem, such as solving both velocity and time in a physics problem. |
| Multiple Answers (MA) | Problems with multiple solutions for a single sub-problem, such as a math fill-in-the-blank problem with answers 1 or -2. |
| Code Generation (CODE) | Problems where the answer is a piece of code, requiring the generation of functional code snippets or complete programs to solve the given task. |
| Others (OT) | Problems that do not fit into the above categories, such as writing chemical equations or explaining reasons, which require human expert evaluation. |

Table 8: Definitions and examples of five image types in our multi-modal scientific problems.

| Image Type | Definition |
| --- | --- |
| Geometric and Mathematical Diagrams | Includes diagrams representing mathematical concepts, such as 2D and 3D shapes, mathematical notations, function plots. |
| Statistical and Data Representation | Visualizations for statistical or data information, including multivariate plots, tables, charts (histograms, bar charts, line plots), and infographics. |
| Natural and Environmental Images | Images of natural scenes or phenomena, including environmental studies visualizations, geological and geographical maps, and satellite images. |
| Scientific and Technical Diagrams | Diagrams used in science, such as cell structures and genetic diagrams in Biology, molecular structures and reaction pathways in Chemistry, force diagrams, circuit diagrams, and astrophysical maps in Physics and Astronomy. |
| Abstract and Conceptual Visuals | Visuals explaining theories and concepts, including flowcharts, algorithms, logic models, and symbolic diagrams. |

# B Data Annotation

## B.1 Problem Extraction and Annotation

We develop a simple and practical annotation interface using Streamlit[12] (as shown in Figure 8). Approximately 30 university students are employed to use this interface for annotation. We provide each annotator with a wage higher than the local average hourly rate. The specific fields annotated are shown in Figure 9. We use image URLs to represent pictures, which allows for efficient storage and easy access without embedding large image files directly in the dataset. Each annotated problem is ultimately stored as a JSON file, facilitating subsequent processing. It is worth mentioning that we embed several rule-based checks and filtering mechanisms in the annotation interface to minimize noise from the annotations. When the following situations arise, we promptly identify and correct the annotations:

1) When the answer type is **Numerical Value**, and the annotated answer contains a variable.

2) When the answer type is not **Numerical Value**, but the annotated answer can be parsed as a numerical value.

3) When the answer type is **Expression**, and the annotated answer contains an equals sign.

4) When the answer type is **Equation**, and the annotated answer does not contain an equals sign.

5) When the annotated answer contains images that should not be present.

6) When the annotated answer contains units (since units are a separate field according to Figure 9, we compile a list of common units and manually check and correct answers when suspected units are detected).

7) When the annotated image links cannot be previewed properly.

Additionally, we implement a multi-step validation process after the initial annotation is completed. First, we conduct a preliminary check using predefined rules to identify any error data, which is then corrected. Following this, a secondary review is performed by different annotators to further check and correct any errors in the annotations. This cross-checking mechanism helps ensure the accuracy and consistency of the annotations.

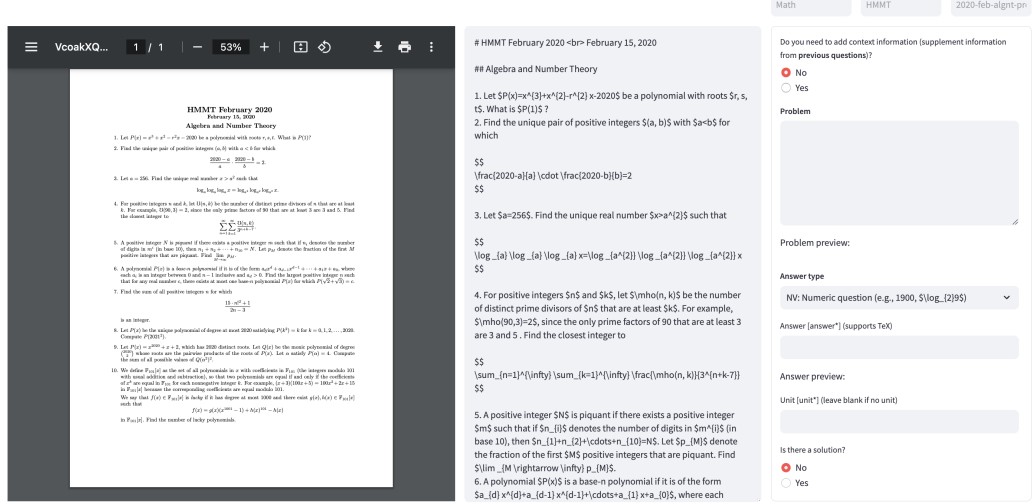

Figure 8: Annotation Page

## B.2 Annotation for Difficulty Levels

The definitions of three levels of difficulty are as follows:

---

[12]https://streamlit.io/

```json
{
    "answer_type": "SC",
    "context": null,
    "problem": "The numbers from 1 to 9 are to be distributed to the nine squares in the diagram according to the following rules:
    There is to be one number in each square. The sum of three adjacent numbers is always a multiple of 3 . The numbers 7 and 9
    are already written in. How many ways are there to insert the remaining numbers?\n\n[figure1]",
    "options": {
        "A": "9",
        "B": "12",
        "C": "15",
        "D": "18",
        "E": "24"
    },
    "answer": "E",
    "unit": null,
    "answer_sequence": null,
    "type_sequence": null,
    "solution": null,
    "figure_urls": {
        "1": {
            "url": "https://i.postimg.cc/mgXQHsKB/image.png",
            "type": "1",
            "caption": "The image displays a simple, black and white representation of a bar graph. The graph consists of a single
            horizontal bar that is divided into two sections. The left section of the bar is labeled with the number \"7\" and the
            right section is labeled with the number \"9\". The numbers are placed inside the respective sections of the bar,
            indicating their values. The bar graph does not contain any additional text or elements. The style of the image is
            minimalistic, with a clear focus on the numerical data represented by the bar."
        }
    },
    "subject": "Math",
    "competition": "MK",
    "file_name": "2023_Student",
    "language": "EN",
    "modality": "multi-modal"
}
```

Figure 9: Example of a json-formatted representation of an annotated problem.

1) **Knowledge Recall:** This involves the direct recall of factual information and well-defined procedures. It examines the memory of simple knowledge points, i.e., whether certain information is known.

2) **Concept Application:** This category covers the very basic use of simple concepts to solve easy problems or perform straightforward calculations. It involves applying known information to situations without any complex or multi-step reasoning. The focus is on straightforward application rather than reasoning.

3) **Cognitive Reasoning:** This involves the use of logical reasoning or visual reasoning to solve problems. It includes problems that require clear thinking and problem-solving techniques. It focuses on the ability to reason and analyze to understand and address the issues.

The prompt we use for categorizing each problem is shown in Figure 10

## B.3   Cognitive Reasoning Abilities Annotation

We provide detailed definitions for each of these cognitive reasoning abilities.

The logical reasoning abilities:

1) **Deductive Reasoning** involves starting with a general principle or hypothesis and logically deriving specific conclusions. This process ensures that the conclusion necessarily follows from the premises.

2) **Inductive Reasoning** involves making broad generalizations from specific observations. This type of reasoning infers general principles from specific instances, enhancing our confidence in the generality of certain phenomena.

3) **Abductive Reasoning** starts with incomplete observations and seeks the most likely explanation. It is used to form hypotheses that best explain the available data.

4) **Analogical Reasoning** involves using knowledge from one situation to solve problems in a similar situation by drawing parallels.

Instructions for Classification: Please classify the above problem by selecting the most appropriate category that best represents the type of thinking and approach required to address the problem. Consider the complexity, the need for creativity, and the depth of knowledge required. You should conclude your response with "So, the problem can be categorized as $\boxed{ANSWER}$.", where ANSWER should be one of the indexes in 1, 2, 3.

Figure 10: The prompt template used for annotating the difficulty level of problems. The "solution" part marked with * is optional.

5) **Cause-and-Effect Reasoning** identifies the reasons behind occurrences and their consequences. This reasoning establishes causal relationships between events.

6) **Critical Thinking** involves objectively analyzing and evaluating information to form a reasoned judgment. It encompasses questioning assumptions and considering alternative explanations.

7) **Decompositional Reasoning** breaks down complex problems or information into smaller, more manageable parts for detailed analysis.

8) **Quantitative Reasoning** involves using mathematical skills to handle quantities and numerical concepts, essential for interpreting data and performing calculations.

The visual reasoning abilities:

1) **Pattern Recognition** is the ability to identify and understand repeating forms, structures, or recurring themes, especially when presented visually. This skill is critical in subjects like Chemistry for recognizing molecular structures, Biology for identifying cellular components, and Geography for interpreting topographic maps.

2) **Spatial Reasoning** is the ability to understand objects in both two and three-dimensional terms and draw conclusions about them with limited information. This skill is often applied in subjects like Math.

Two-Dimensional Examples: Plane geometry, segments, lengths.

Three-Dimensional Examples: Solid geometry, spatial visualization

3) **Diagrammatic Reasoning** represents the capability to solve problems expressed in diagrammatic form, understanding the logical connections between shapes, symbols, and texts.

Examples: Reading various forms of charts and graphs, obtaining and analyzing statistical information from diagrams.

4) **Symbol Interpretation** is the ability to decode and understand abstract and symbolic visual information. Examples: Understanding abstract diagrams, interpreting symbols, including representations of data structures such as graphs and linked lists

5) **Comparative Visualization** represents comparing and contrasting visual elements to discern differences or similarities, often required in problem-solving to determine the relationship between variable components.

The prompt we use for annotating different logical reasoning abilities and visual reasoning abilities are shown separately in Figure 11 and Figure 12.

---

Problem description:
{problem}

Answer:
{answer}

Solution:
{solution}*

You need to identify and select the specific types of logical reasoning abilities required to solve the question from the list provided below.

Logical Reasoning Abilities:

1. Deductive Reasoning: Deductive reasoning involves starting with a general principle or hypothesis and logically deriving specific conclusions. This process ensures that the conclusion necessarily follows from the premises.
2. Inductive Reasoning: Inductive reasoning involves making broad generalizations from specific observations. This type of reasoning infers general principles from specific instances, enhancing our confidence in the generality of certain phenomena.
3. Abductive Reasoning: Abductive reasoning starts with incomplete observations and seeks the most likely explanation. It is used to form hypotheses that best explain the available data.
4. Analogical Reasoning: Analogical reasoning involves using knowledge from one situation to solve problems in a similar situation by drawing parallels.
5. Cause-and-Effect Reasoning: Cause-and-effect reasoning identifies the reasons behind occurrences and their consequences. This reasoning establishes causal relationships between events.
6. Critical Thinking: Critical thinking involves objectively analyzing and evaluating information to form a reasoned judgment. It encompasses questioning assumptions and considering alternative explanations.
7. Decompositional Reasoning: Decompositional reasoning breaks down complex problems or information into smaller, more manageable parts for detailed analysis.
8. Quantitative Reasoning: Quantitative reasoning involves using mathematical skills to handle quantities and numerical concepts, essential for interpreting data and performing calculations.

Analyze the question, its answer and explanation (if provided) to determine which of the above reasoning abilities are necessary. Conclude by clearly stating which reasoning abilities are involved in solving the question using "So, the involved reasoning abilities are $\boxed{ABILITIES}$", where "ABILITIES" represents the numbers corresponding to the list above, separated by commas if multiple abilities are relevant.

---

Figure 11: The prompt template used for annotating different logical reasoning abilities of problems. The "solution" part marked with * is optional.

Problem description:
{problem}

Answer:
{answer}

Solution:
{solution}*

You need to identify and select the specific types of visual reasoning abilities required to solve the question from the list provided below.

Visual Reasoning Abilities:

1. Pattern Recognition: The ability to identify and understand repeating forms, structures, or recurring themes, especially when presented visually. This skill is critical in subjects like Chemistry for recognizing molecular structures, Biology for identifying cellular components, and Geography for interpreting topographic maps.
2. Spatial Reasoning: Spatial reasoning is the ability to understand objects in both two and three-dimensional terms and draw conclusions about them with limited information. This skill is often applied in subjects like Math. Two-Dimensional Examples: Plane geometry, segments, lengths Three-Dimensional Examples: Solid geometry, spatial visualization
3. Diagrammatic Reasoning: The capability to solve problems expressed in diagrammatic form, understanding the logical connections between shapes, symbols, and texts. Examples: Reading various forms of charts and graphs, obtaining and analyzing statistical information from diagrams
4. Symbol Interpretation: The ability to decode and understand abstract and symbolic visual information. Examples: Understanding abstract diagrams, interpreting symbols, including representations of data structures such as graphs and linked lists
5. Comparative Visualization: Comparing and contrasting visual elements to discern differences or similarities, often required in problem-solving to determine the relationship between variable components.

Analyze the question, its answer, and any explanation provided to determine which of the above reasoning abilities are necessary. Conclude by clearly stating which reasoning abilities are involved in solving the question using "So, the involved reasoning abilities are $\boxed{ABILITIES}$", where "ABILITIES" represents the numbers corresponding to the list above, separated by commas if multiple abilities are relevant.

Figure 12: The prompt template used for annotating different visual reasoning abilities of problems which have multi-modal inputs. The "solution" part marked with * is optional.

## C Experiment Details

### C.1 Prompt for Image Caption

The prompt we use for captioning each image in the benchmark for LMMs is shown in Figure 13.

### C.2 Models

In our experiments, we evaluate a range of both open-source and proprietary LMMs and LLMs. For LMMs, we select the newly released GPT-4o [36] and the powerful GPT-4V [1] from OpenAI. Additionally, we include Claude3 Sonnet [3] from Anthropic, and Gemini Pro Vision[13] [45] from Google, and Qwen-VL-Max [6] from Alibaba. We also evaluate several open-source models, includ-

---

[13]We do not test Gemini-1.5-pro [39] as there are significant rate limits on accessing the model's API during the time we do experiments.

> [Image]
>
> Describe the fine-grained content of the image or figure, including scenes, objects, relationships, and any text present.

Figure 13: The prompt template used for image caption.

Table 9: LMMs and their corresponding LLMs.

| LMM | LLM |
| --- | --- |
| GPT-4o | GPT-4o |
| GPT-4v | GPT-4 |
| Claude3 Sonnet | Claude3 Sonnet |
| Gemini Pro Vision | Gemini Pro |
| LLaVA-NeXT-34B | Nous-Hermes-2-Yi-34B |
| InternVL-Chat-V1.5 | InternLM2-20B-Chat |
| Yi-VL-34B | Yi-34B-Chat |
| Qwen-VL-Chat | Qwen-7B-Chat |

ing LLaVA-NeXT-34B [31], InternVL-Chat-V1.5 [12], Yi-VL-34B [55], and Qwen-VL-Chat [7]. For LLMs, we primarily select the corresponding text models of the aforementioned LMMs, such as GPT-4 [2]. Additionally, we include open-source models like Qwen-7B-Chat, Qwen1.5-32B-Chat [5], Yi-34B-Chat [55], and InternLM2-Chat-20B [8]. Table 9 shows the relationship between LMMs and their corresponding LLMs. For the proprietary models, we call the APIs, while for the open-source models, we run them on an 8-card A800 cluster.

## C.3 Evaluation Prompts

We meticulously design the prompts used for model input during experiments. These prompts are tailored to different answer types, with specific output formats specified for each type. The detailed prompt templates are shown in Figure 14, and the different instructions for each answer type are provided in Table 10.

> You are participating in an international {subject} competition and need to solve the following question.
>
> {answer type description}
>
> Here is some context information for this question, which might assist you in solving it: {context}*
>
> Problem:
> {problem}
>
> All mathematical formulas and symbols you output should be represented with LaTeX. You can solve it step by step and please end your response with: {answer format instruction}.

Figure 14: The prompt template used for problem input. The "context" part marked with * is optional and refers to supplementary information provided during manual annotation when the problem relies on conclusions from previous questions. The {answer type description} and {answer format instruction} are specified in Table 10.

## C.4 Model Hyperparameters

For all models, we set the maximum number of output tokens to 2048 and the temperature to 0.0. When performing code generation **(CODE)** tasks, the temperature is set to 0.2.

## C.5 Answer-level Evaluation Protocols

**Rule-based Evaluation**    For problems with fixed answers, we extract the final answer enclosed in "\boxed{}" (using prompts to instruct models to conclude their final answers with boxes) and perform rule-based matching according to the answer type.

1) For numerical value **(NV)** answers, we handle units by explicitly stating them in the prompts provided to the model, if applicable. During evaluation, we assess only the numerical value output by the model, disregarding the unit. In cases where numerical answers are subject to estimation, such as in physics or chemistry problems, we convert both the model's output and the correct answer to scientific notation. If the exponent of 10 is the same for both, we allow a deviation of 0.1 in the coefficient before the exponent, accounting for minor estimation errors in the model's calculations.

2) For problems where the answer type is an expression **(EX)** or an equation **(EQ)**, we use the SymPy[14] library for comparison. This allows us to accurately assess the equivalence of algebraic expressions and equations by symbolic computation.

3) For problems requiring the solution of multiple quantities **(MPV)**, our evaluation strictly follows the order of output specified in the prompt, ensuring consistency and correctness in the sequence of results.

4) In the case of problems with multiple answers **(MA)**, we require the model to output all possible answers, adequately considering various scenarios.

5) For problems where the answer type is an interval **(IN)**, we strictly compare the open and closed intervals as well as the boundary values of the endpoints.

6) For problems where the answer type is a set **(SET)**, we compare the set output by the model with the standard answer set to ensure they are completely identical. For problems where the answer type is a tuple **(TUP)**, we compare the tuple output by the model with the standard answer tuple to ensure that each corresponding position is exactly equal.

7) For code generation **(CODE)** problems, we extract the code output by the model and test it through all provided test cases. Specifically, we use the unbiased pass@k metric,

$$\text{pass}@k := \mathop{\mathbb{E}}_{\text{Problems}} \left[ 1 - \frac{\binom{n-c}{k}}{\binom{n}{k}} \right] \tag{1}$$

where we set $k = 1$ and $n = 5$, and $c$ indicates the number of correct samples that pass all test cases.

**Model-based Evaluation**    To deal with those problems with answer types that cannot be appropriately evaluated using rule-based matching, we employ model-based evaluation. In this approach, we utilize GPT-4V as the evaluator. We design prompts that include the problem, the correct answer, the solution (if provided), and the response from the model being tested (see Figure 15 for details). The evaluator model then judges the correctness of the tested model's response.

To further ensure the reliability of using a model as an evaluator, we uniformly sampled 100 problems across various subjects that involved model evaluation. We have several students with backgrounds in science and engineering independently conduct manual evaluations. It turns out that out of the 100 sampled problems, there is nearly $80\%$ agreement between the human evaluations and the model evaluations. Considering that problems requiring model-based evaluation account for approximately 5% of the total, the error rate can be controlled at around $20\% \times 5\%$, which is approximately $1\%$. Therefore, we consider this method to be reliable.

---

[14]https://www.sympy.org/

> You are an experienced teacher tasked with grading an Olympic-level {subject} exam paper.
>
> The problem's context:
> {context}*
>
> Problem:
> {problem}
>
> The student's answer:
> {the tested model's response}
>
> The reference answer:
> {the reference answer}
>
> The reference solution:
> {the reference solution}*
>
> Note:
> (1) You can tolerate some markdown formatting issues.
> (2) You need to make judgments based on the provided reference answer and reference solution (if provided).
> You can analyze the answer step by step, and then output $\boxed{correct}$ or $\boxed{incorrect}$ at the end to express your final judgment.

Figure 15: The prompt used for model-based evaluation. The "context" and "the reference solution" parts marked with * are optional.

## C.6 Process-level Evaluation Protocols

To conduct the process-level evaluation, we utilize a method based on GPT-4V. First, we reformat both the gold solution and the model-generated solution for the sampled problems into a neat step-by-step format using GPT-4. Then, we employ a carefully designed prompt(see Figure 16) to guide GPT-4V using the reformatted gold solution to evaluate the correctness of each step in the model's output, assigning a score of 0 for incorrect and 1 for correct steps. The final process-level score for each problem is determined by averaging the scores of all the steps.

## D  Fine-grained Results

### D.1  Results across Logical and Visual Reasoning Abilities

Table 11 and Table 12 show the performance of different models across various logical and visual reasoning abilities separately.

### D.2  Results on Multimodal Problems

Table 13 shows the performance of different models on multimodal problems across different subjects.

### D.3  Process-level Evaluation Results

Table 14 shows process-level results of different models across different subjects.

### D.4  Results across Different Languages

Table 15 shows results of different models in different languages.

You are a teacher skilled in evaluating the intermediate steps of a student's solution to a given problem.

You are given two types of step-by-step solutions: one from the reference answer and the other from the student. Your task is to evaluate the correctness of each step in the student's solutions using binary scoring: assign a score of 1 for correct steps and 0 for incorrect steps. Use the reference solutions to guide your evaluation.
Follow the format:
Step 1: ...
Step 2: ...
Step 3: ...
Please provide the results directly, omitting any introductory or concluding remarks.
# The given question

{the question}

# The reference solution

{the reference solution}

# The student's solution

{the model's solution}

# Your scores for each step of the student's solutions

Figure 16: The prompt used for process-level evaluation.

## D.5 Error Analysis

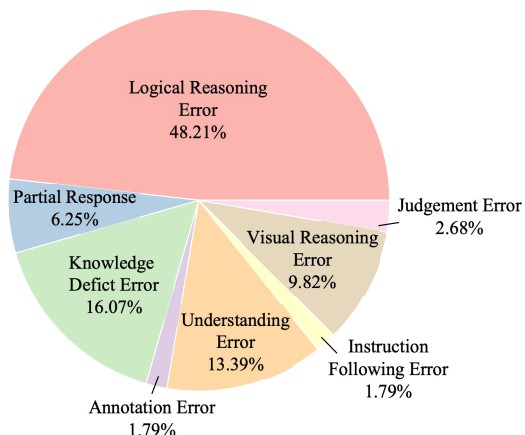

Figure 17: The distribution of error types of 112 sampled problems where GPT-4V makes errors.

We sample incorrect responses from GPT-4V (16 problems per subject, with 8 text-only and 8 multimodal) and have human evaluators analyze and annotate the reasons for these errors. As depicted in Figure 17, reasoning errors (both logical and visual) constitute the largest category, indicating that our benchmark effectively highlights the current models' deficiencies in cognitive reasoning abilities. Additionally, a significant portion of errors stem from knowledge deficits, suggesting that current models still lack expert-level domain knowledge and the ability to leverage this knowledge to assist in reasoning. Another category of errors arise from understanding biases, which can be attributed to

the models' misinterpretation of context and difficulties in integrating complex language structures and multimodal information. More relevant cases are shown in Appendix F.1.

# E Data Leakage Detection Details

We combine the questions and detailed solutions (or answers if there are no steps) of the problems, then use the n-gram prediction accuracy metric. Specifically, for each sample, we sample k starting points and predict the next 5-gram each time. To evaluate whether the n-gram prediction is correct, we use exact match and more lenient metrics such as edit distance and ROUGE-L. Here, we consider a prediction correct if either the edit distance or ROUGE-L similarity exceeds 75%, to mitigate some reformatting issues during pre-training. We take the union of instances detected by different metrics to obtain the final set of detected instances.

As shown in Tables 16, 17, and 18, the experimental results reveal that indeed, different models exhibit minor leakage across different subjects. An interesting observation is that some leakages detected by the base model are no longer detectable when using the chat model based on the same base model. We hypothesize that optimization for dialogue capabilities potentially impacts the model's ability and performance on the next token prediction. Another similar observation is that leakages detected by text-only chat models tend to decrease when evaluated on multimodal chat models based on the same chat models. Figure 18 presents a data leakage case from Qwen1.5-32B-Chat.

Figure 18: A potential data leakage case of Qwen1.5-32B-Chat which is presented with the original problem and solution concatenated, separated by a space.

Table 10: Descriptions of answer types and corresponding format instructions included in the problem input prompts. Specifically, {unit description} indicates: "Remember, your answer should be calculated in the unit of {unit}, but do not include the unit in your final answer."

| Answer Type | Answer Type Description | Answer Format Instruction |
|---|---|---|
| SC | This is a multiple choice question (only one correct answer). | Please end your response with: "The final answer is $\boxed{ANSWER}$", where ANSWER should be one of the options: {the options of the problem}. |
| MC | This is a multiple choice question (more than one correct answer). | Please end your response with: "The final answer is $\boxed{ANSWER}$", where ANSWER should be two or more of the options: {the options of the problem}. |
| TF | This is a True or False question. | Please end your response with: "The final answer is $\boxed{ANSWER}$", where ANSWER should be either "True" or "False". |
| NV | The answer to this question is a numerical value. | {unit instruction} Please end your response with: "The final answer is $\boxed{ANSWER}$", where ANSWER is the numerical value without any units. |
| SET | The answer to this question is a set. | {unit instruction} Please end your response with: "The final answer is $\boxed{ANSWER}$", where ANSWER is the set of all distinct answers, each expressed as a numerical value without any units, e.g. ANSWER = {3, 4, 5}. |
| IN | The answer to this question is a range interval. | {unit instruction} Please end your response with: "The final answer is $\boxed{ANSWER}$", where ANSWER is an interval without any units, e.g. ANSWER = $(1,2] \cup [7, +\infty)$. |
| EX | The answer to this question is an expression. | {unit instruction} Please end your response with: "The final answer is $\boxed{ANSWER}$", where ANSWER is an expression without any units and equals signs, e.g. ANSWER = $\frac{1}{2}gt^2$. |
| EQ | The answer to this question is an equation. | {unit instruction} Please end your response with: "The final answer is $\boxed{ANSWER}$", where ANSWER is an equation without any units, e.g. ANSWER = $\frac{x^2}{4} + \frac{y^2}{2} = 1$. |
| TUP | The answer to this question is a tuple. | {unit instruction} Please end your response with: "The final answer is $\boxed{ANSWER}$", where ANSWER is a tuple without any units, e.g. ANSWER=(3, 5). |
| MPV | This question involves multiple quantities to be determined. | Your final quantities should be output in the following order: {the ordered sequence of the name of multiple quantities}. Their units are, in order, {the ordered sequence of the units}, but units shouldn't be included in your concluded answer. Their answer types are, in order, {the ordered sequence of answer types}. Please end your response with: "The final answers are $\boxed{ANSWER}$", where ANSWER should be the sequence of your final answers, separated by commas, for example: 5, 7, 2.5. |
| MA | This question has more than one correct answer, you need to include them all. | Their units are, in order, {the ordered sequence of the units}, but units shouldn't be included in your concluded answer. Their answer types are, in order, {the ordered sequence of answer types}. Please end your response with: "The final answers are $\boxed{ANSWER}$", where ANSWER should be the sequence of your final answers, separated by commas, for example: 5, 7, 2.5. |
| CODE | Write a Python program to solve the given competitive programming problem using standard input and output methods. Pay attention to time and space complexities to ensure efficiency. | Notes: (1) Your solution must handle standard input and output. Use `input()` for reading input and `print()` for output. (2) Be mindful of the problem's time and space complexity. The solution should be efficient and designed to handle the upper limits of input sizes within the given constraints. (3) It's encouraged to analyze and reason about the problem before coding. You can think step by step, and finally output your final code in the following format: `Your Python code here` |
| OT | - | - |

Table 11: Experimental results across different logical reasoning abilities on OlympicArena benchmark, expressed as percentages, with the highest score in each setting underlined and the highest scores across all settings bolded. **DED**: Deductive Reasoning, **IND**: Inductive Reasoning, **ABD**: Abductive Reasoning, **ANA**: Analogical Reasoning, **CAE**: Cause-and-Effect Reasoning, **CT**: Critical Thinking, **DEC**: Decompositional Reasoning, **QUA**: Quantitative Reasoning.

| Model | DED | IND | ABD | ANA | CAE | CT | DEC | QUA |
|---|---|---|---|---|---|---|---|---|
| | Accuracy | Accuracy | Accuracy | Accuracy | Accuracy | Accuracy | Accuracy | Accuracy |
| LLMs | | | | | | | | |
| Qwen-7B-Chat | 4.85 | 4.18 | 4.84 | 5.29 | 5.54 | 5.16 | 4.09 | 4.64 |
| Yi-34B-Chat | 19.65 | 13.84 | 26.82 | 18.73 | 26.51 | 25.71 | 15.00 | 15.55 |
| Internlm2-20B-Chat | 17.43 | 13.12 | 24.74 | 16.30 | 22.81 | 22.51 | 13.03 | 13.42 |
| Qwen1.5-32B-Chat | 25.94 | 21.20 | 33.39 | 24.87 | 32.33 | 31.82 | 20.19 | 22.19 |
| GPT-3.5 | 19.38 | 13.19 | 26.64 | 16.30 | 23.32 | 24.31 | 14.43 | 17.35 |
| Claude3 Sonnet | 25.40 | 17.88 | 34.78 | 23.28 | 30.64 | 31.15 | 18.59 | 22.67 |
| GPT-4 | 33.93 | 24.80 | 40.66 | 33.33 | 38.48 | 39.32 | 26.84 | 31.72 |
| GPT-4o | 39.1 | 30.14 | 43.43 | 37.78 | 42.89 | 44.06 | 31.79 | 36.56 |
| Image caption + LLMs | | | | | | | | |
| Qwen-7B-Chat | 5.66 | 4.69 | 7.27 | 6.88 | 6.66 | 6.02 | 4.60 | 4.81 |
| Yi-34B-Chat | 19.08 | 13.34 | 29.24 | 20.11 | 25.53 | 24.91 | 13.79 | 14.64 |
| Internlm2-20B-Chat | 18.25 | 12.69 | 28.37 | 17.67 | 23.84 | 23.35 | 13.40 | 14.52 |
| Qwen1.5-32B-Chat | 25.50 | 20.55 | 35.81 | 26.35 | 31.35 | 31.39 | 19.55 | 21.51 |
| GPT-3.5 | 20.71 | 13.91 | 29.76 | 17.78 | 25.72 | 26.01 | 15.74 | 17.73 |
| Claude3 Sonnet | 25.69 | 19.11 | 35.12 | 24.02 | 30.88 | 31.55 | 18.71 | 22.89 |
| GPT-4 | 35.06 | 24.44 | 41.35 | 34.39 | 40.17 | 40.72 | 27.26 | 32.47 |
| GPT-4o | 39.26 | 30.93 | 45.50 | 39.37 | 43.17 | 44.19 | 31.35 | 36.56 |
| LMMs | | | | | | | | |
| Qwen-VL-Chat | 7.87 | 6.06 | 12.80 | 8.68 | 9.90 | 9.92 | 5.29 | 6.29 |
| Yi-VL-34B | 16.30 | 10.60 | 21.11 | 16.40 | 21.35 | 20.76 | 11.89 | 13.42 |
| InternVL-Chat-V1.5 | 17.65 | 12.55 | 30.28 | 17.25 | 22.06 | 22.70 | 12.56 | 14.37 |
| LLaVA-NeXT-34B | 19.72 | 14.70 | 30.62 | 19.37 | 27.40 | 25.39 | 13.62 | 14.79 |
| Qwen-VL-Max | 22.97 | 16.87 | 33.91 | 21.38 | 29.28 | 28.51 | 17.26 | 18.13 |
| Gemini Pro Vision | 22.45 | 17.16 | 35.47 | 21.59 | 25.67 | 27.54 | 17.24 | 18.91 |
| Claude3 Sonnet | 25.59 | 18.89 | 36.51 | 23.70 | 29.89 | 31.71 | 18.99 | 22.87 |
| GPT-4V | 34.59 | 25.59 | 46.54 | 33.33 | 39.61 | 41.15 | 26.47 | 30.79 |
| GPT-4o | **41.18** | **32.73** | **50.35** | **40.53** | **45.94** | **47.12** | **33.17** | **37.58** |

Table 12: Experimental results across different visual reasoning abilities on OlympicArena benchmark, expressed as percentages, with the highest score in each setting underlined and the highest scores across all settings bolded. **PR**: Pattern Recognition, **SPA**: Spatial Reasoning, **DIA**: Diagrammatic Reasoning, **SYB**: Symbol Interpretation, **COM**: Comparative Visualization.

| Model | PR | SPA | DIA | SYB | COM |
|---|---|---|---|---|---|
| | Accuracy | Accuracy | Accuracy | Accuracy | Accuracy |
| LLMs | | | | | |
| Qwen-7B-Chat | 4.59 | 2.64 | 4.26 | 4.01 | 4.66 |
| Yi-34B-Chat | 23.70 | 13.58 | 19.56 | 17.61 | 22.37 |
| Internlm2-20B-Chat | 22.89 | 13.06 | 18.63 | 15.73 | 21.16 |
| Qwen1.5-32B-Chat | 28.93 | 17.94 | 24.67 | 22.18 | 27.83 |
| GPT-3.5 | 22.33 | 13.27 | 18.40 | 16.05 | 21.05 |
| Claude3 Sonnet | 26.88 | 17.60 | 22.86 | 20.49 | 25.98 |
| GPT-4 | 33.65 | 23.99 | 30.09 | 27.94 | 32.54 |
| GPT-4o | 35.96 | 28.71 | 33.29 | 31.54 | 35.00 |
| Image caption + LLMs | | | | | |
| Qwen-7B-Chat | 5.96 | 4.11 | 5.21 | 5.11 | 6.30 |
| Yi-34B-Chat | 21.69 | 21.19 | 18.01 | 14.92 | 20.48 |
| Internlm2-20B-Chat | 22.97 | 12.75 | 18.27 | 15.49 | 21.05 |
| Qwen1.5-32B-Chat | 28.59 | 17.81 | 23.90 | 20.95 | 26.73 |
| GPT-3.5 | 23.96 | 15.26 | 19.72 | 17.34 | 22.30 |
| Claude3 Sonnet | 27.60 | 17.03 | 22.84 | 20.17 | 26.28 |
| GPT-4 | 34.29 | 26.07 | 31.07 | 28.61 | 33.11 |
| GPT-4o | 37.08 | 29.10 | 33.60 | 31.22 | 35.91 |
| LMMs | | | | | |
| Qwen-VL-Chat | 9.90 | 4.93 | 7.46 | 6.48 | 8.91 |
| Yi-VL-34B | 16.72 | 9.60 | 13.78 | 12.10 | 15.09 |
| InternVL-Chat-V1.5 | 22.85 | 12.11 | 17.68 | 15.11 | 21.27 |
| LLaVA-NeXT-34B | 24.69 | 12.75 | 19.72 | 16.38 | 22.90 |
| Qwen-VL-Max | 27.43 | 16.26 | 22.35 | 19.47 | 26.01 |
| Gemini Pro Vision | 28.98 | 14.83 | 21.65 | 19.79 | 26.13 |
| Claude3 Sonnet | 27.18 | 17.55 | 22.43 | 20.84 | 25.56 |
| GPT-4V | 35.28 | 23.91 | 30.25 | 27.70 | 34.40 |
| GPT-4o | **41.49** | **30.65** | **36.98** | **33.91** | **40.58** |

Table 13: Experimental results on multimodal problems on OlympicArena benchmark, expressed as percentages, with the highest score in each setting underlined and the highest scores across all settings bolded.

| Model | Math | Physics | Chemistry | Biology | Geography | Astronomy | CS | Overall |
|---|---|---|---|---|---|---|---|---|
| | Accuracy | Accuracy | Accuracy | Accuracy | Accuracy | Accuracy | Pass@1 | Accuracy |
| LLMs | | | | | | | | |
| Qwen-7B-Chat | 1.26 | 2.54 | 6.92 | 5.59 | 4.16 | 2.70 | 0 | 4.01 |
| Yi-34B-Chat | 5.04 | 6.14 | 19.15 | 27.40 | 33.91 | 10.00 | 0.28 | 19.54 |
| Internlm2-20B-Chat | 6.30 | 6.46 | 15.76 | 26.96 | 31.87 | 9.19 | 0.97 | 18.51 |
| Qwen1.5-32B-Chat | 7.98 | 8.90 | 23.86 | 32.21 | 39.74 | 18.65 | 0.83 | 24.58 |
| GPT-3.5 | 6.30 | 7.20 | 15.46 | 26.85 | 30.66 | 11.62 | 6.25 | 18.79 |
| Claude3 Sonnet | 8.82 | 11.76 | 19.59 | 31.99 | 38.00 | 15.68 | 2.64 | 23.79 |
| GPT-4 | 16.81 | 18.43 | 32.11 | 39.71 | 41.26 | 23.92 | 12.50 | 31.05 |
| GPT-4o | 21.85 | 21.82 | 32.11 | 42.17 | 44.28 | 30.68 | 12.78 | 34.11 |
| Image caption + LLMs | | | | | | | | |
| Qwen-7B-Chat | 3.78 | 2.22 | 6.19 | 6.49 | 7.34 | 5.00 | 0 | 5.32 |
| Yi-34B-Chat | 5.04 | 6.57 | 14.29 | 24.94 | 33.61 | 8.78 | 0.28 | 18.25 |
| Internlm2-20B-Chat | 6.72 | 6.89 | 16.35 | 25.39 | 31.26 | 10.27 | 1.18 | 18.41 |
| Qwen1.5-32B-Chat | 6.72 | 8.69 | 21.80 | 32.10 | 39.82 | 17.30 | 0.97 | 23.99 |
| GPT-3.5 | 4.20 | 12.39 | 18.11 | 25.73 | 32.32 | 9.73 | 7.64 | 20.04 |
| Claude3 Sonnet | 5.46 | 13.35 | 20.47 | 32.89 | 38.23 | 13.38 | 3.89 | 23.97 |
| GPT-4 | 16.81 | 20.44 | 29.90 | 38.7 | 45.12 | 26.62 | 12.26 | 32.36 |
| GPT-4o | 21.01 | 22.14 | 31.22 | 45.19 | 45.19 | 30.54 | **14.58** | 34.86 |
| LMMs | | | | | | | | |
| Qwen-VL-Chat | 3.36 | 2.65 | 6.63 | 9.84 | 13.85 | 5.27 | 0 | 7.82 |
| Yi-VL-34B | 3.36 | 6.46 | 9.13 | 18.79 | 22.03 | 7.43 | 0 | 13.00 |
| InternVL-Chat-V1.5 | 7.56 | 6.25 | 16.05 | 24.94 | 32.55 | 9.73 | 0.62 | 18.43 |
| LLaVA-NeXT-34B | 4.62 | 6.46 | 14.43 | 28.30 | 36.11 | 10.00 | 0.28 | 19.66 |
| Qwen-VL-Max | 6.30 | 7.63 | 17.82 | 28.86 | 40.05 | 15.14 | 1.25 | 22.38 |
| Gemini Pro Vision | 7.56 | 9.11 | 24.30 | 32.55 | 35.81 | 11.22 | 2.36 | 22.58 |
| Claude3 Sonnet | 5.46 | 13.45 | 19.15 | 33.22 | 37.02 | 17.30 | 2.36 | 24.05 |
| GPT-4V | 13.87 | 18.22 | 29.31 | 40.27 | 46.86 | 22.43 | 11.25 | 31.81 |
| GPT-4o | **26.47** | **22.14** | **33.14** | **46.98** | **53.75** | **31.76** | 13.61 | **38.17** |

Table 14: Results of the process-level evaluation on our comprehensive OlympicArena benchmark. Each step of every problem is assigned a score of 0 (indicating incorrect) or 1 (indicating correct), with the highest score in each setting underlined and the highest scores across all settings highlighted in bold. The subject of computer science is neglected in this part due to the lack of solutions.

| Model | Math | Physics | Chemistry | Biology | Geography | Astronomy | Overall |
|-------|------|---------|-----------|---------|-----------|-----------|---------|
| | Score | Score | Score | Score | Score | Score | Score |
| LLMs | | | | | | | |
| Qwen-7B-Chat | 18.7 | 43.7 | 35.1 | 18.9 | 34.5 | 31.5 | 30.4 |
| Yi-34B-Chat | 30.2 | 51.0 | 54.0 | 31.9 | 36.5 | 40.3 | 40.7 |
| Internlm2-20B-Chat | 21.2 | 35.0 | 51.2 | 22.7 | 32.9 | 33.3 | 32.7 |
| Qwen1.5-32B-Chat | 32.0 | 44.0 | 61.1 | 32.0 | 45.2 | 48.6 | 43.8 |
| GPT-3.5 | 37.6 | 46.9 | 32.7 | 30.2 | 38.7 | 26.7 | 35.4 |
| Claude3 Sonnet | 40.8 | 42.7 | 65.3 | 30.8 | 52.6 | 50.5 | 47.1 |
| GPT-4 | 57.0 | 53.8 | 73.6 | 50.0 | 50.1 | 65.0 | 58.2 |
| GPT-4o | 59.9 | **65.9** | 67.4 | 49.6 | **61.4** | 69.5 | 62.3 |
| Image caption + LLMs | | | | | | | |
| Qwen-7B-Chat | 23.0 | 42.6 | 34.6 | 17.4 | 34.4 | 32.3 | 30.7 |
| Yi-34B-Chat | 26.3 | 45.6 | 49.5 | 20.0 | 45.7 | 42.0 | 38.2 |
| Internlm2-20B-Chat | 27.7 | 42.6 | 46.3 | 19.4 | 25.5 | 43.1 | 34.1 |
| Qwen1.5-32B-Chat | 35.9 | 49.7 | 56.8 | 33.5 | 43.6 | 51.4 | 45.1 |
| GPT-3.5 | 32.1 | 46.7 | 51.2 | 29.1 | 38.4 | 38.2 | 39.3 |
| Claude3 Sonnet | 50.7 | 51.7 | 66.1 | 33.4 | 55.8 | 52.2 | 51.7 |
| GPT-4 | 61.4 | 53.8 | 62.7 | 51.1 | 52.0 | 62.2 | 57.2 |
| GPT-4o | 54.3 | 63.3 | 71.8 | **58.6** | 56.6 | 72.6 | **62.9** |
| LMMs | | | | | | | |
| Qwen-VL-Chat | 14.3 | 41.7 | 35.7 | 21.0 | 31.0 | 23.6 | 27.9 |
| Yi-VL-34B | 28.9 | 41.0 | 44.2 | 18.7 | 30.2 | 40.3 | 33.9 |
| InternVL-Chat-V1.5 | 26.6 | 40.5 | 42.7 | 29.4 | 43.1 | 44.8 | 37.8 |
| LLaVA-NeXT-34B | 30.2 | 47.1 | 50.1 | 19.0 | 40.6 | 47.1 | 39.0 |
| Qwen-VL-Max | 27.5 | 52.4 | 65.5 | 24.3 | 36.0 | 48.4 | 42.3 |
| Gemini Pro Vision | 28.5 | 46.4 | 45.2 | 19.9 | 33.5 | 40.5 | 35.7 |
| Claude3 Sonnet | 47.3 | 46.8 | 63.2 | 24.2 | 43.2 | 48.1 | 45.5 |
| GPT-4V | 49.9 | 54.0 | 71.1 | 51.4 | 56.3 | 64.3 | 57.8 |
| GPT-4o | **60.2** | 54.8 | **72.2** | 51.6 | 59.6 | **74.4** | 62.1 |

Table 15: Experimental results across different languages (English and Chinese) on OlympicArena benchmark, expressed as percentages, with the highest score in each setting underlined and the highest scores across all settings bolded.

| Model | English Accuracy | Chinese Accuracy |
|---|---|---|
| LLMs | | |
| Qwen-7B-Chat | 4.17 | 4.55 |
| Yi-34B-Chat | 16.37 | 18.89 |
| Internlm2-20B-Chat | 16.56 | 16.62 |
| Qwen1.5-32B-Chat | 22.73 | 25.29 |
| GPT-3.5 | 19.83 | 15.50 |
| Claude3 Sonnet | 25.73 | 18.20 |
| GPT-4 | 35.13 | 27.31 |
| GPT-4o | 40.65 | 33.66 |
| Image caption + LLMs | | |
| Qwen-7B-Chat | 4.71 | 5.21 |
| Yi-34B-Chat | 16.96 | 16.26 |
| Internlm2-20B-Chat | 17.40 | 16.43 |
| Qwen1.5-32B-Chat | 22.93 | 24.24 |
| GPT-3.5 | 20.56 | 15.77 |
| Claude3 Sonnet | 26.31 | 17.43 |
| GPT-4 | 36.08 | 27.40 |
| GPT-4o | 41.50 | 33.07 |
| LMMs | | |
| Qwen-VL-Chat | 7.70 | 5.55 |
| Yi-VL-34B | 17.34 | 14.68 |
| InternVL-Chat-V1.5 | 17.07 | 15.82 |
| LLaVA-NeXT-34B | 17.74 | 16.74 |
| Qwen-VL-Max | 20.14 | 21.49 |
| Gemini Pro Vision | 21.61 | 18.76 |
| Claude3 Sonnet | 26.52 | 17.21 |
| GPT-4V | 36.18 | 26.55 |
| GPT-4o | **43.04** | **34.39** |

Table 16: Full results of Data Leakage Detection on the base models or text-only chat models behind the evaluated models (continued). The "Correspondence" column indicates the text-only chat model and multimodal (MM) chat model corresponding to the model being detected. "# Leak." denotes the number of leakage instances. "# T" represents the number of instances correctly answered among these leaks by the text-only chat model, while "# MM" represents the number of instances correctly answered among these leaks by the multimodal chat model.

| Model to-be-detected | Correspondence | | Math | | | Physics | | | Chemistry | | |
|---|---|---|---|---|---|---|---|---|---|---|---|
| | Text-only Chat Model | MM Chat Model | # Leak. | # T | # MM | # Leak. | # T | # MM | # Leak. | # T | # MM |
| InternLM2-20B | InternLM2-20B-Chat | InternVL-Chat-1.5 | 14 | 1 | 2 | 3 | 0 | 0 | 0 | 0 | 0 |
| internLM2-20B-Chat | InternLM2-20B-Chat | InternVL-Chat1.5 | 17 | 1 | 0 | 0 | 0 | 0 | 1 | 1 | 1 |
| Yi-34B | Yi-34B-Chat | Yi-VL-34B | 10 | 2 | 2 | 1 | 0 | 0 | 0 | 0 | 0 |
| Yi-34B-Chat | Yi-34B-Chat | Yi-VL-34B | 2 | 0 | 0 | 0 | 0 | 0 | 0 | 0 | 0 |
| Nous-Hermes-2-Yi-34B | - | LLaVA-NeXT-34B | 0 | - | 0 | 0 | - | 0 | 0 | - | 0 |
| Qwen-7B | Qwen-7B-Chat | Qwen-VL-Chat | 8 | 0 | 0 | 1 | 1 | 0 | 0 | 0 | 0 |
| Qwen1.5-32B | Qwen1.5-32B-Chat | - | 24 | 3 | - | 3 | 1 | - | 1 | 0 | - |
| Qwen1.5-32B-Chat | Qwen1.5-32B-Chat | - | 19 | 2 | - | 4 | 2 | - | 3 | 1 | - |
| GPT-4o | GPT-4o | GPT-4o | 0 | 0 | 0 | 0 | 0 | 0 | 0 | 0 | 0 |

Table 17: Full results of Data Leakage Detection on the base models or text-only chat models behind the evaluated models (continued). The "Correspondence" column indicates the text-only chat model and multimodal (MM) chat model corresponding to the model being detected. "# Leak." denotes the number of leakage instances. "# T" represents the number of instances correctly answered among these leaks by the text-only chat model, while "# MM" represents the number of instances correctly answered among these leaks by the multimodal chat model.

| Model to-be-detected | Correspondence | | Biology | | | Geography | | | Astronomy | | |
|---|---|---|---|---|---|---|---|---|---|---|---|
| | Text-only Chat Model | MM Chat Model | # Leak. | # T | # MM | # Leak. | # T | # MM | # Leak. | # T | # MM |
| InternLM2-20B | InternLM2-20B-Chat | InternVL-Chat-1.5 | 0 | 0 | 0 | 0 | 0 | 0 | 1 | 0 | 0 |
| InternLM2-20B-Chat | InternLM2-20B-Chat | InternVL-Chat-1.5 | 0 | 0 | 0 | 0 | 0 | 0 | 0 | 0 | 0 |
| Yi-34B | Yi-34B-Chat | Yi-VL-34B | 1 | 0 | 0 | 0 | 0 | 0 | 0 | 0 | 0 |
| Yi-34B-Chat | Yi-34B-Chat | Yi-VL-34B | 0 | 0 | 0 | 0 | 0 | 0 | 0 | 0 | 0 |
| Nous-Hermes-2-Yi-34B | - | LLaVA-NeXT-34B | 0 | - | 0 | 0 | - | 0 | 0 | - | 0 |
| Qwen-7B | Qwen-7B-Chat | Qwen-VL-Chat | 0 | 0 | 0 | 0 | 0 | 0 | 1 | 0 | 0 |
| Qwen1.5-32B | Qwen1.5-32B-Chat | - | 1 | 0 | - | 0 | 0 | - | 5 | 1 | - |
| Qwen1.5-32B-Chat | Qwen1.5-32B-Chat | - | 1 | 0 | - | 0 | 0 | - | 0 | 0 | - |
| GPT-4o | GPT-4o | GPT-4o | 0 | 0 | 0 | 0 | 0 | 0 | 0 | 0 | 0 |

Table 18: Full results of Data Leakage Detection on the base models or text-only chat models behind the evaluated models (continued). The "Correspondence" column indicates the text-only chat model and multimodal (MM) chat model corresponding to the model being detected. "# Leak." denotes the number of leakage instances. "# T" represents the number of instances correctly answered among these leaks by the text-only chat model, while "# MM" represents the number of instances correctly answered among these leaks by the multimodal chat model.

| Model to-be-detected | Correspondence | | CS | | | Overall | | |
|---|---|---|---|---|---|---|---|---|
| | Text-only Chat Model | MM Chat Model | # Leak. | # T | # MM | # Leak. | # T | # MM |
| InternLM2-20B | InternLM2-20B-Chat | InternVL-Chat-1.5 | 1 | 1 | 1 | 19 | 2 | 3 |
| InternLM2-20B-Chat | InternLM2-20B-Chat | InternVL-Chat-1.5 | 1 | 1 | 1 | 19 | 3 | 2 |
| Yi-34B | Yi-34B-Chat | Yi-VL-34B | 0 | 0 | 0 | 12 | 2 | 2 |
| Yi-34B-Chat | Yi-34B-Chat | Yi-VL-34B | 0 | 0 | 0 | 2 | 0 | 0 |
| Nous-Hermes-2-Yi-34B | - | LLaVA-NeXT-34B | 0 | - | 0 | 0 | 0 | 0 |
| Qwen-7B | Qwen-7B-Chat | Qwen-VL-Chat | 1 | 1 | 1 | 11 | 2 | 1 |
| Qwen1.5-32B | Qwen1.5-32B-Chat | - | 9 | - | - | 43 | 14 | 0 |
| Qwen1.5-32B-Chat | Qwen1.5-32B-Chat | - | 3 | 3 | - | 30 | 8 | 0 |
| GPT-4o | GPT-4o | GPT-4o | 0 | 0 | 0 | 0 | 0 | 0 |

# F Case Study

## F.1 Cases for Error Analysis

From Figure 19 to Figure 25, we showcase examples of various error types across different disciplines.

## Math - Logical Reasoning Error

**Problem:**
In the diagram, rectangle $PQRS$ is placed inside rectangle $ABCD$ in two different ways: first, with $Q$ at $B$ and $R$ at $C$; second, with $P$ on $AB$, $Q$ on $BC$, $R$ on $CD$, and $S$ on $DA$. [figure1]. If $AB = 718$ and $PQ = 250$, determine the length of $BC$.

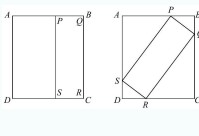

[figure1]

**Solution:**
Let $BC = x$, $PB = b$, and $BQ = a$. Since $BC = x$, then $AD = PS = QR = x$. Since $BC = x$ and $BQ = a$, then $QC = x - a$. Since $AB = 718$ and $PB = b$, then $AP = 718 - b$. Note that $PQ = SR = 250$. Let $\angle BQP = \theta$. Since $\triangle PBQ$ is right-angled at $B$, then $\angle BPQ = 90° - \theta$. Since $BQC$ is a straight angle and $\angle PQR = 90°$, then $\angle RQC = 180° - 90° - \theta = 90° - \theta$. Since $APB$ is a straight angle and $\angle SPQ = 90°$, then $\angle APS = 180° - 90° - (90° - \theta) = \theta$. Since $\triangle SAP$ and $\triangle QCR$ are each right-angled and have another angle in common with $\triangle PBQ$, then these three triangles are similar. Continuing in the same way, we can show that $\triangle RDS$ is also similar to these three triangles. Since $RS = PQ$, then $\triangle RDS$ is actually congruent to $\triangle PBQ$ (angle-side-angle). Similarly, $\triangle SAP$ is congruent to $\triangle QCR$. In particular, this means that $AS = x - a$, $SD = a$, $DR = b$, and $RC = 718 - b$. Since $\triangle SAP$ and $\triangle PBQ$ are similar, then $\frac{SA}{PB} = \frac{AP}{BQ} = \frac{SP}{PQ}$. Thus, $\frac{x-a}{b} = \frac{718-b}{a} = \frac{x}{250}$. Also, by the Pythagorean Theorem in $\triangle PBQ$, we obtain $a^2 + b^2 = 250^2$. By the Pythagorean Theorem in $\triangle SAP$,

$$x^2 = (x - a)^2 + (718 - b)^2$$
$$x^2 = x^2 - 2ax + a^2 + (718 - b)^2$$
$$0 = -2ax + a^2 + (718 - b)^2$$

Since $a^2 + b^2 = 250^2$, then $a^2 = 250^2 - b^2$. Since $\frac{718-b}{a} = \frac{x}{250}$, then $ax = 250(718 - b)$. Therefore, substituting into ($*$), we obtain

$$0 = -2(250)(718 - b) + 250^2 - b^2 + (718 - b)^2$$
$$b^2 = 250^2 - 2(250)(718 - b) + (718 - b)^2$$
$$b^2 = ((718 - b) - 250)^2 \quad \left(\text{since } y^2 - 2yz + z^2 = (y - z)^2\right)$$
$$b^2 = (468 - b)^2$$
$$b = 468 - b \quad (\text{since } b \neq b - 468)$$
$$2b = 468$$
$$b = 234$$

Therefore, $a^2 = 250^2 - b^2 = 250^2 - 234^2 = (250 + 234)(250 - 234) = 484 \cdot 16 = 22^2 \cdot 4^2 = 88^2$ and so $a = 88$.
Finally, $x = \frac{250(718-b)}{a} = \frac{250 \cdot 484}{88} = 1375$. Therefore, $BC = \boxed{1375}$.

**GPT 4V's output:**
To solve this problem, we can use the Pythagorean theorem. In the first placement of rectangle $PQRS$ inside rectangle $ABCD$, we see that $PQ$ is parallel to $AB$ and $QR$ is parallel to $BC$. Since $PQ = 250$ and $AB = 718$, we can denote the length of $BC$ as $x$. In the second placement, the rectangle $PQRS$ is rotated such that its corners touch the sides of rectangle $ABCD$. The diagonal $PR$ of rectangle $PQRS$ becomes the diameter of an inscribed circle within rectangle $ABCD$. This means that the diagonal $PR$ is the hypotenuse of a right triangle with sides $PQ$ and $QR$. Since $PQRS$ is a rectangle, $PQ = RS$ and $PS = QR$. Therefore, the length of $QR$ is also $x$. Now we have a right triangle with sides 250 and $x$, and the hypotenuse is the diagonal $PR$. Using the Pythagorean theorem, we have:

$$PR^2 = PQ^2 + QR^2$$
$$PR^2 = 250^2 + x^2$$
$$PR^2 = 62500 + x^2$$

The diagonal $PR$ also bisects rectangle $ABCD$ into two right triangles, each with sides $AB$ and $BC$. Therefore, the length of the diagonal $PR$ can also be calculated using the Pythagorean theorem with sides $AB$ and $BC$:

$$PR^2 = AB^2 + BC^2$$
$$PR^2 = 718^2 + x^2$$
$$PR^2 = 515524 + x^2$$

Since both expressions represent $PR^2$, we can set them equal to each other:

$$62500 + x^2 = 515524 + x^2$$

Subtract $x^2$ from both sides:

$$62500 = 515524$$

Now we see that there is a mistake in the calculation because the left side should not equal the right side directly. We need to subtract 62500 from 515524 to find the value of $x^2$:

$$x^2 = 515524 - 62500$$
$$x^2 = 453024$$

Now we take the square root of both sides to find $x$:

$$x = \sqrt{453024}$$
$$x = 673$$

Therefore, the length of $BC$ is 673. The final answer is $\boxed{673}$.

Figure 19: An example of a math problem with a logical reasoning error.

## Physics - Logical Reasoning Error

**Problem:**
A bead is placed on a horizontal rail, along which it can slide frictionlessly. It is attached to the end of a rigid, massless rod of length $R$. A ball is attached at the other end. Both the bead and the ball have mass $M$. The system is initially stationary, with the ball directly above the bead. The ball is then given an infinitesimal push, parallel to the rail.[figure1] Assume that the rod and ball are designed in such a way (not shown explicitly in the diagram) so that they can pass through the rail without hitting it. In other words, the rail only constrains the motion of the bead. Two subsequent states of the system are shown below.[figure2] Derive an expression for the force in the rod when the ball is directly below the bead, as shown at right above.

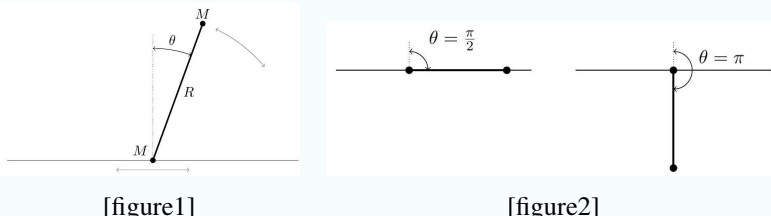

[figure1]                                    [figure2]

**Solution:**
At this point the released gravitational potential energy is $2MgR$, and both masses are moving horizontally with speed $v$, where

$$2\frac{1}{2}Mv^2 = 2MgR \quad \Rightarrow \quad v = \sqrt{2gR}$$

Work in the frame moving to the right with speed $v$. In this frame the bead is stationary and the ball has velocity $2v$ and is instantaneously rotating about the bead, so it must be experiencing a centripetal force

$$\frac{M(2v)^2}{R} = 8Mg$$

Unlike in part (a), there are no additional contribution from the acceleration of the rotation center, because the bead can only ever accelerate horizontally, and the force in the rod at this moment is vertical. Since the ball also experiences a downward force of $Mg$ due to gravity, the force in the rod is a tension

$$T = \boxed{9Mg}$$

**GPT 4V's output:**
To derive an expression for the force in the rod when the ball is directly below the bead, we can employ the conservation of energy and principles from circular motion.

Since the bead and ball are on a frictionless rail and assuming no other non-conservative forces are acting on the system, the total mechanical energy is conserved. The initial potential energy of the ball when it is given an infinitesimal push will convert into kinetic energy as the ball swings down.

At the top position (initial state), the potential energy (PE) is maximum and kinetic energy (KE) is zero. We define the gravitational potential energy to be zero at the lowest point. At the highest point, the potential energy is $MgR$ because the height of the ball above the lowest point is $R$.

Hence, $PE_{\text{initial}} = MgR$.

At the bottom position (final state), when the ball is directly below the bead, all the potential energy will have been converted to kinetic energy (assuming conservation of energy). If $v$ is the velocity of the ball at the bottom, then $KE_{\text{final}} = \frac{1}{2}Mv^2$.

Since energy is conserved, $PE_{\text{initial}} = KE_{\text{final}}$. Thus,

$$MgR = \frac{1}{2}Mv^2$$
$$2gR = v^2$$
$$v = \sqrt{2gR}$$

Now, let us consider the forces acting on the ball at the bottom position. We have the gravitational force $Mg$ acting downwards and the force in the rod $F$ acting upwards. When the ball is at the bottom, it is in circular motion and the centripetal force required to keep it in circular motion is provided entirely by the force in the rod, which is the centripetal force $F_{\text{centripetal}}$, since gravity acts in the same direction as the centripetal acceleration at this point.

The centripetal force is given by $F_{\text{centripetal}} = \frac{Mv^2}{R}$.

Substituting $v = \sqrt{2gR}$ into the above equation, we get

$$F_{\text{centripetal}} = \frac{M(2gR)}{R}$$
$$F_{\text{centripetal}} = 2Mg$$

Since the centripetal force is provided by the force in the rod and directed towards the center of the circular motion (upwards towards the ball), at this moment, the rod is under tension, not compression.

Thus, the force in the rod when the ball is directly below the bead and in tension is $F = 2Mg$.

The final answer is $\boxed{2Mg}$.

Figure 20: An example of a physics problem with a logical reasoning error.

**Problem:**

At temperature $t^\circ$C, a solution is prepared such that c$(H_2CO_3)$ + c$\left(HCO_3^-\right)$ + c$\left(CO_3^{2-}\right)$ = $1.000 \times 10^{-3}$ mol $\cdot$ L$^{-1}$. This solution is a mixture of $H_2CO_3$ and HCl or $H_2CO_3$ and NaOH. The negative logarithms of the concentrations of some particles in the solution ($-$ lgc) versus pH are shown in the figure. Which of the following statements are incorrect:

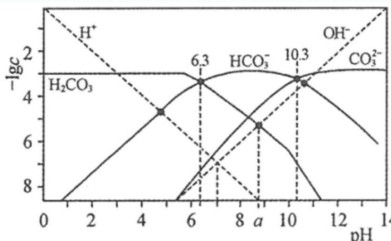

[figure1]

A: In the solution with pH = a: c$\left(HCO_3^-\right)$ > c$(H_2CO_3)$ > c$\left(CO_3^{2-}\right)$ > c$\left(H^+\right)$

B: In the solution with pH = 7: c$\left(Na^+\right)$ > c$(H_2CO_3)$

C: In the solution with pH = 10.3: c$\left(Na^+\right)$ < $1.000 \times 10^{-3}$ mol $\cdot$ L$^{-1}$

D: At $25^\circ$C, the equilibrium constant for the reaction $H_2CO_3 + CO_3^{2-} \rightleftharpoons 2HCO_3^-$ is $1.0 \times 10^4$

**Solution:**

A. According to the information from the figure, in the solution with pH = a, c$\left(CO_3^{2-}\right)$ > c$(H_2CO_3)$, so statement A is incorrect.

B. In the solution with pH = 7, according to the information from the figure, c$\left(HCO_3^-\right)$ > c$(H_2CO_3)$ > c$\left(CO_3^{2-}\right)$. According to charge conservation and pH = 7, c$\left(Na^+\right)$ = c$\left(HCO_3^-\right)$ + 2c$\left(CO_3^{2-}\right)$, so c$\left(Na^+\right)$ > c$(H_2CO_3)$. Therefore, statement B is correct.

C. At $25^\circ$C, when pH = 6.3, c$(H_2CO_3)$ = c$\left(HCO_3^-\right)$, then $K_a$ $(H_2CO_3)$ = c$\left(H^+\right)$ = $1.000 \times 10^{-6.3}$ mol $\cdot$ L$^{-1}$. In the solution with pH = 10.3, c$\left(CO_3^{2-}\right)$ = c$\left(HCO_3^-\right)$, $K_a$ $\left(HCO_3^-\right)$ = c$\left(H^+\right)$ = $1.000 \times 10^{-10.3}$ mol $\cdot$ L$^{-1}$. Then $K_a$ $(H_2CO_3)$ = $\frac{c\left(HCO_3^-\right) \times c\left(H^+\right)}{c(H_2CO_3)}$ = $\frac{c\left(HCO_3^-\right) \times 10^{-10.3}}{c(H_2CO_3)}$ = $10^{-6.3}$. When pH = 10.3, c$\left(CO_3^{2-}\right)$ = c$\left(HCO_3^-\right)$, so $\frac{c\left(HCO_3^-\right) + c\left(CO_3^{2-}\right)}{c(H_2CO_3)}$ = $2 \times 10^4$, c$(H_2CO_3)$ + c$\left(HCO_3^-\right)$ + c$\left(CO_3^{2-}\right)$ = $1.000 \times 10^{-3}$ mol $\cdot$ L$^{-1}$. Therefore, c$(H_2CO_3)$ = $5.000 \times 10^{-8}$ mol $\cdot$ L$^{-1}$, and 2c$\left(CO_3^{2-}\right)$ = $1.000 \times 10^{-3}$ mol $\cdot$ L$^{-1}$ $-$ $5.000 \times 10^{-8}$ mol $\cdot$ L$^{-1}$. According to charge conservation, c$\left(Na^+\right)$ > c$\left(HCO_3^-\right)$ + 2c$\left(CO_3^{2-}\right)$, i.e., c$\left(Na^+\right)$ > 3c$\left(CO_3^{2-}\right)$, and c$\left(Na^+\right)$ > $1.000 \times 10^{-3}$ mol $\cdot$ L$^{-1}$. Therefore, statement C is incorrect.

D. At $25^\circ$C, when pH = 6.3, c$(H_2CO_3)$ = c$\left(HCO_3^-\right)$, then $K_a$ $(H_2CO_3)$ = c$\left(H^+\right)$ = $1.000 \times 10^{-6.3}$ mol $\cdot$ L$^{-1}$. Similarly, at pH = 10.3, c$\left(CO_3^{2-}\right)$ = c$\left(HCO_3^-\right)$, then $K_a$ $\left(HCO_3^-\right)$ = c$\left(H^+\right)$ = $1.000 \times 10^{-10.3}$ mol $\cdot$ L$^{-1}$. The equilibrium constant for the reaction $H_2CO_3 + CO_3^{2-} \rightleftharpoons 2HCO_3^-$ is:

$$K = \frac{c^2\left(HCO_3^-\right)}{c(H_2CO_3) \times c\left(CO_3^{2-}\right)} = \frac{c\left(HCO_3^-\right) \times c\left(HCO_3^-\right) \times c\left(H^+\right)}{c(H_2CO_3) \times c\left(CO_3^{2-}\right) \times c(H^+)} = \frac{K_a \left(H_2CO_3\right)}{K_a \left(HCO_3^-\right)} = \frac{10^{-6.3}}{10^{-10.3}} = 1 \times 10^4$$

Therefore, statement D is correct.

In conclusion, the answer is $\boxed{A, C}$.

**GPT 4V's output:**

First, we need to analyze the chart, which shows the variation of the negative logarithm of concentrations (-lgc) of $H_2CO_3$, $HCO_3^-$, $CO_3^{2-}$, and $H^+$ at different pH values. This chart is a typical acid-base titration curve where $H_2CO_3$ is a diprotic weak acid that can dissociate into $HCO_3^-$ and $CO_3^{2-}$.

Option A: At pH = a, according to the chart, the concentration of $HCO_3^-$ is the highest, followed by $H_2CO_3$, then $CO_3^{2-}$, and the concentration of $H^+$ is the lowest. This is consistent with the chart.

Option B: At pH = 7, the concentration of $H_2CO_3$ should be relatively low because it is a weak acid and does not dissociate completely at neutral pH. The concentration of $Na^+$ depends on whether NaOH has been added. If NaOH has been added, the concentration of $Na^+$ will increase. However, this option does not provide enough information to determine the exact concentration relationship between $Na^+$ and $H_2CO_3$, so we cannot confirm if this option is correct.

Option C: At pH = 10.3, according to the chart, the concentration of $CO_3^{2-}$ is the highest, and the concentration of $HCO_3^-$ is very low at this pH. If NaOH is added to the solution, the concentration of $Na^+$ will increase. However, since the total concentration $c(H_2CO_3) + c(HCO_3^-) + c(CO_3^{2-})$ = $1.000 \times 10^{-3}$ mol$\cdot$L$^{-1}$, the concentration of $Na^+$ cannot be lower than this value. Therefore, this option is incorrect.

Option D: The equilibrium constant K for the reaction $H_2CO_3 + CO_3^{2-} \rightleftharpoons 2HCO_3^-$ can be expressed by the following equation:

$$K = \frac{[HCO_3^-]^2}{[H_2CO_3][CO_3^{2-}]}$$

According to the chart, at pH = 10.3, the concentrations of $HCO_3^-$ and $CO_3^{2-}$ are equal, which means $K = 1.0 \times 10^4$ is impossible because this would require $[HCO_3^-]$ to be much greater than $[CO_3^{2-}]$. Therefore, this option is also incorrect.

In summary, options C and D are incorrect. So the final answer is $\boxed{C, D}$

Figure 21: An example of a chemistry problem with a visual reasoning error.

## Biology - Logical Reasoning Error

**Problem:**

The bacterium Bradyrhizobium japonicum can infect soybean (Glycine max) roots and form nodules. The nitrogen fixation catalyzed by nitrogenase occurs in the nodules and the nitrogenase activity can be measured easily by acetylene reduction instead of nitrogen reduction. Scientists generated a defective mutation of $NAD^+$-dependent malic enzyme, the enzyme that generates pyruvate and $NADH$, and infected soybean seedling roots with wildtype and mutant bacteria. The seedlings were grown in nitrogen-free media. After 14 and 28 days of inoculation, the number and weight of nodules in the seedlings and acetylene reduction activity were recorded [figure1]. Which of the following statements are correct:

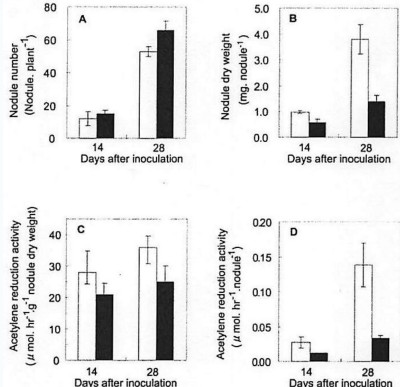

[figure1]: Nodule number and dry weight and acetylene reduction acitivity of soybean. Soybean nodules infected with wild-type B. japonicum (open bars) and the dme mutant (solid bars) are presented.

A: Nitrogen fixation activity in nodules of the same treament at 28 days after inoculation is higher than that at 14 days after inoculation.
B: Both number and size of nodules increase with time from 14 to 28 days after inoculation with $B$. japonicum.
C: The reduction in nitrogen-fixing activity of nodules infected by the mutant at 28 days after inoculation compared to those at 14 days after inoculation is due to the reduction of nitrogenase activity and nodule formation.
D: Nitrogen fixation in B. japonicum -induced nodule is down-regulated by $NAD^+$ dependent malic enzyme.

**Solution:**

A. True. As shown in the figure, in both wildtype and mutant treaments, acetylene reduction activity in nodules at 28 days after inoculation is higher than that at 14 days after inoculation, indicating higher nitrogen fixation actitity.
B. True. In both treaments, number and size of nodules are higher at 28 days compared to those at 14 days after inoculation
C. False. Number of nodules at 28 days after inoculation is higher than at 14 days after inoculation
D. False. The mutation of $NAD^+$-dependent malic enzyme in the bacteria results in a reduction of acetylene reduction, indicating that the enzyme (in wildtype) up-regulates the nitrogen fixation.

So we choose $\boxed{A, B}$.

**GPT 4V's output:**

Let's analyze the data presented in Fig.Q13 step by step:

A: We need to determine if nitrogen fixation activity in nodules of the same treatment at 28 days after inoculation is higher than that at 14 days after inoculation. Looking at the graphs C and D, which represent acetylene reduction activity (a proxy for nitrogen fixation activity), we can see that for both wild-type (open bars) and mutant (solid bars) B. japonicum, the activity is higher at 28 days than at 14 days. This is true for both per gram of nodule dry weight (graph C) and per nodule (graph D). Therefore, statement A is correct.

B: To assess this statement, we need to look at graphs A and B. Graph A shows the nodule number per plant, and graph B shows the nodule dry weight. For both wild-type and mutant B. japonicum, the number and size (weight) of nodules increase from 14 to 28 days after inoculation. Therefore, statement B is correct.

C: This statement suggests that the reduction in nitrogen-fixing activity of nodules infected by the mutant at 28 days after inoculation compared to those at 14 days is due to the reduction of nitrogenase activity and nodule formation. From graph C, we can see that the acetylene reduction activity per gram of nodule dry weight for the mutant does decrease from 14 to 28 days, indicating a reduction in nitrogenase activity. However, graph A shows that the number of nodules actually increases for the mutant from 14 to 28 days, which contradicts the second part of the statement. Therefore, statement C is not entirely correct.

D: The statement suggests that nitrogen fixation in B. japonicum-induced nodules is down-regulated by NAD+-dependent malic enzyme. The mutant, which has a defective mutation in the NAD+-dependent malic enzyme, shows lower acetylene reduction activity compared to the wild-type in both graphs C and D at both time points. This suggests that the enzyme plays a role in nitrogen fixation, and its absence (or defectiveness) leads to down-regulation of nitrogen fixation. Therefore, statement D is correct.

The final answer is $\boxed{A, B, D}$.

Figure 22: An example of a biology problem with a logical reasoning error.

## Geography - Knowledge Deficit Error

**Problem:**
From December 17 to 20, 2020, the Beijing team went to Tengchong, Yunnan to participate in the study tour course design competition, using the Beihai Wetland in Tengchong, known for its plateau volcanic dammed lakes, as the theme of the study tour course design. Based on this, complete the following small questions. During the study tour, the team members found that compared to Beijing, Tengchong has ( ).

A. The tree shadows at sunrise are more towards the north.
B. The temperature difference between day and night is larger.
C. The sunset time is earlier.
D. The tree shadows at noon are shorter.

**Solution:**
According to the material, Tengchong $(25^\circ\,\mathrm{N}, 98^\circ\,\mathrm{E})$ belongs to Yunnan and is located to the southwest of Beijing $(39^\circ\,\mathrm{N}, 116^\circ\,\mathrm{E})$. From December 17 to 20, 2020, it was winter, around the winter solstice, with the sun rising in the southeast and the shadow at sunrise being in the northwest. Beijing is at a higher latitude than Tengchong, causing a greater shadow deviation, so A is incorrect. Tengchong is located near a plateau volcanic dammed lake, rich in geothermal energy, resulting in high nighttime temperatures and smaller temperature differences between day and night, so B is incorrect. Tengchong, being at a lower latitude, has longer daylight hours and a later sunset time, making C incorrect. Tengchong's lower latitude also means a larger solar altitude angle at noon, leading to shorter tree shadows, so $\boxed{D}$ is correct.

**GPT 4V's output:**
First, we need to understand the geographical locations of Beijing and Tengchong. Beijing is in the northern part of China, while Tengchong is in the southwestern part of China, near the Myanmar border. This means that Tengchong is to the southwest of Beijing.
A. The tree shadows at sunrise are more towards the north - Due to the Earth's rotation, the direction of tree shadows at sunrise is mainly determined by latitude, not by the east-west position. Therefore, this option is incorrect.
B. The temperature difference between day and night is larger - Tengchong is located in a plateau area, while Beijing is in a plain area. Generally, the temperature difference between day and night in plateau areas is larger than in plain areas because high-altitude areas cool down quickly at night and receive strong solar radiation during the day. Therefore, this option might be correct.
C. The sunset time is earlier - The sunset time is mainly determined by longitude, as the Earth rotates from west to east. Tengchong is to the southwest of Beijing, but this does not necessarily mean that the sunset time is earlier. This option requires more information to determine.
D. The tree shadows at noon are shorter - The solar altitude angle at noon depends on latitude; the lower the latitude, the higher the solar altitude angle, and the shorter the tree shadows. Tengchong has a lower latitude than Beijing, so the tree shadows at noon should be shorter.
In summary, options B and D are both possible. However, we need to determine which one is most related to the "plateau volcanic dammed lake" mentioned in the problem. Since this description emphasizes the plateau characteristics of Tengchong, we can reasonably infer that the larger temperature difference between day and night is most relevant to this characteristic. Therefore, the final answer is $\boxed{B}$.

Figure 23: An example of a geography problem with a knowledge deficit error.

## Astronomy - Incomplete Response

**Problem:**

In order to move a spacecraft between orbits we must apply a thrust using rockets, which changes the velocity of the spacecraft by $\Delta v$. In this question we will ignore changes in the mass of the spacecraft due to the burning of fuel. For an object of mass $m$ in a circular orbit of radius $r$ around an object with mass $M$ (where $m \ll M$ ) the orbital velocity, $v_{\text{orb}}$ , is given by the formula $v_{\text{orb}} = \sqrt{\frac{GM}{r}}$. Approximating Mars' orbit as circular with a radius of $1.52 \text{AU}$, calculate the $\Delta v$ to go from Earth LEO to Mars i.e. $\Delta v = |\Delta v_A| + |\Delta v_B|$. Compare your answer to the $\Delta v$ to reach Earth LEO.

**Solution:**

$$\Delta v_{\text{A}} = \sqrt{\frac{6.67 \times 10^{-11} \times 1.99 \times 10^{30}}{1.49 \times 10^{11}}} \left( \sqrt{\frac{2 \times 1.52}{1 + 1.52}} - 1 \right)$$

$$= 2935 \text{ m s}^{-1} \left( = 2.94 \text{ km s}^{-1} \right)$$

$$\Delta v_{\text{B}} = \sqrt{\frac{6.67 \times 10^{-11} \times 1.99 \times 10^{30}}{1.52 \times 1.49 \times 10^{11}}} \left( 1 - \sqrt{\frac{2 \times 1}{1 + 1.52}} \right)$$

$$= 2642 \text{ m s}^{-1} \left( = 2.64 \text{ km s}^{-1} \right)$$

$$\Delta v = 2.94 + 2.64$$

$$= \boxed{5.58} \text{ km s}^{-1}$$

[The $\Delta v$ calculated here would be for the spacecraft to enter a circular orbit around the Sun at the same distance as Mars, but this would not constitute landing - the extra $\Delta v$ to get to the Martian surface increases the total for the whole transfer to roughly $8 \text{ km s}^{-1}$ (same as to get into LEO). Other, more complicated routes can be taken, some of which offer substantial efficiencies so the $\Delta v$ for the trip can be much lower (although the time taken to complete the manoeuvre will be longer, and the $\Delta v$ to reach LEO will still be the biggest single step)]

**GPT 4V's output:**

To calculate the $\Delta v$ to go from Earth Low Earth Orbit (LEO) to Mars, we need to consider two main maneuvers: the transfer from Earth's orbit to the transfer orbit (Hohmann transfer orbit), and then from the transfer orbit to Mars' orbit.

1. The first part of the $\Delta v$, denoted as $\Delta v_A$, is the change in velocity needed to move from Earth's orbit to the transfer orbit. This is the velocity at perihelion of the transfer orbit minus the Earth's orbital velocity.

2. The second part, $\Delta v_B$, is the change in velocity needed to move from the transfer orbit to Mars' orbit. This is Mars' orbital velocity minus the velocity at aphelion of the transfer orbit.

The Hohmann transfer orbit is an elliptical orbit that touches Earth's orbit at its closest point to the Sun (perihelion) and Mars' orbit at its farthest point from the Sun (aphelion).

The semi-major axis of the transfer orbit ($a_t$) is the average of the Earth's orbital radius ($r_E$) and Mars' orbital radius ($r_M$). Given that 1 AU (Astronomical Unit) is approximately $1.496 \times 10^8$ km, we have:

$r_E = 1 \times 1.496 \times 10^8$ km (Earth's orbit radius) $r_M = 1.52 \times 1.496 \times 10^8$ km (Mars' orbit radius)

$a_t = \frac{r_E + r_M}{2}$

The orbital velocity of an object is given by $v_{\text{orb}} = \sqrt{\frac{GM}{r}}$, where $G$ is the gravitational constant and $M$ is the mass of the central object (in this case, the Sun).

The velocity at perihelion of the transfer orbit ($v_{p,t}$) is given by:

$v_{p,t} = \sqrt{\frac{GM}{r_E}} \times \sqrt{\frac{2r_M}{r_E + r_M}}$

The velocity at aphelion of the transfer orbit ($v_{a,t}$) is given by:

$v_{a,t} = \sqrt{\frac{GM}{r_M}} \times \sqrt{\frac{2r_E}{r_E + r_M}}$

The Earth's orbital velocity ($v_E$) is:

$v_E = \sqrt{\frac{GM}{r_E}}$

Mars' orbital velocity ($v_M$) is:

$v_M = \sqrt{\frac{GM}{r_M}}$

Now we can calculate $\Delta v_A$ and $\Delta v_B$:

$\Delta v_A = v_{p,t} - v_E$ $\Delta v_B = v_M - v_{a,t}$

And the total $\Delta v$ is:

$\Delta v = |\Delta v_A| + |\Delta v_B|$

Let's calculate these values step by step. We will use the gravitational constant $G = 6.674 \times 10^{-20}$ km$^3$kg$^{-1}$s$^{-2}$ and the mass of the Sun $M = 1.989 \times 10^{30}$ kg.

First, we calculate the semi-major axis of the transfer orbit:

$a_t = \frac{1 \times 1.496 \times 10^8 + 1.52 \times 1.496 \times 10^8}{2}$

Now, we calculate the velocities:

$v_{p,t} = \sqrt{\frac{6.674 \times 10^{-20} \times 1.989 \times 10^{30}}{1.496 \times 10^8}} \times \sqrt{\frac{2 \times 1.52 \times 1.496 \times 10^8}{1.496 \times 10^8 + 1.52 \times 1.496 \times 10^8}}$

$v_{a,t} = \sqrt{\frac{6.674 \times 10^{-20} \times 1.989 \times 10^{30}}{1.52 \times 1.496 \times 10^8}} \times \sqrt{\frac{2 \times 1.496 \times 10^8}{1.496 \times 10^8 + 1.52 \times 1.496 \times 10^8}}$

$v_E = \sqrt{\frac{6.674 \times 10^{-20} \times 1.989 \times 10^{30}}{1.496 \times 10^8}}$

$v_M = \sqrt{\frac{6.674 \times 10^{-20} \times 1.989 \times 10^{30}}{1.52 \times 1.496 \times 10^8}}$

Now we can calculate $\Delta v_A$ and $\Delta v_B$:

$\Delta v_A = v_{p,t} - v_E$

$\Delta v_B = v_M - v_{a,t}$

And finally, the total $\Delta v$:

$\Delta v = |\Delta v_A| + |\Delta v_B|$

Let's calculate these values.

Figure 24: An example of an astronomy problem with an incomplete response.

## Computer Science - Understanding Error

**Problem:**

A girl named Xenia has a cupboard that looks like an arc from ahead. The arc is made of a semicircle with radius r (the cupboard's top) and two walls of height h (the cupboard's sides). The cupboard's depth is r, that is, it looks like a rectangle with base r and height h+r from the sides. The figure below shows what the cupboard looks like (the front view is on the left, the side view is on the right). Xenia got lots of balloons for her birthday. The girl hates the mess, so she wants to store the balloons in the cupboard. Luckily, each balloon is a sphere with radius. Help Xenia calculate the maximum number of balloons she can put in her cupboard. [figure1] You can say that a balloon is in the cupboard if you can't see any part of the balloon on the left or right view. The balloons in the cupboard can touch each other. It is not allowed to squeeze the balloons or deform them in any way. You can assume that the cupboard's walls are negligibly thin.

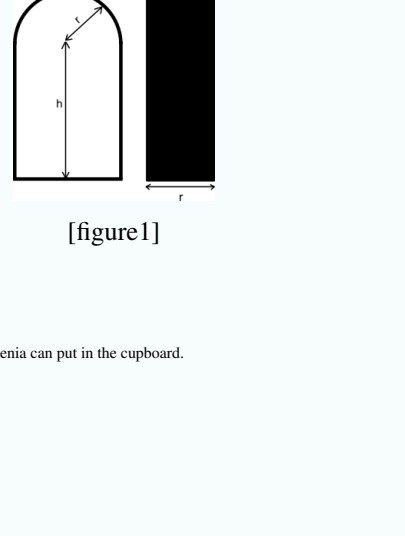

[figure1]

Input
The single line contains two integers r, h ($1 <= r, h <= 10^7$)

Output
Print a single integer — the maximum number of balloons Xenia can put in the cupboard.

Examples

Input
1 1

Output
3

Input
1 2

Output
5

Input
2 1

Output
2

time limit: 2 seconds
memory limit: 256 megabytes

**GPT 4V's output:**

```python
import math

# Function to calculate the maximum number of balloons
def max_balloons(r, h):
    # Calculate the number of balloons that fit in the height of the walls
    balloons_in_height = (2 * h) // r

    # Calculate the number of balloons that fit in the semicircle on top
    # The diameter of the semicircle is r, so it can fit one balloon of radius r/2
    # The remaining height in the semicircle is r/2, which can fit another balloon
    balloons_in_semicircle = 1 + (h * 2 + r) // r

    # Calculate the number of additional balloons that can fit in the corners of the semicircle
    # This is the number of full r/2 radii that fit into the remaining arc length after placing
    # the first balloon in the semicircle
    remaining_arc_length = (math.pi * r / 2) - (r / 2)
    additional_balloons = int(remaining_arc_length // (r / 2))

    # The total number of balloons is the sum of the above calculated values
    total_balloons = balloons_in_height + balloons_in_semicircle + additional_balloons

    return total_balloons

# Read input
r, h = map(int, input().split())

# Calculate and print the result
print(max_balloons(r, h))
```

This program defines a function "max balloons" that calculates the maximum number of balloons that can fit in the cupboard based on the given dimensions. It then reads the input values for "r" and "h", calls the function with these values, and prints the result. The program takes into account the balloons that fit in the height of the walls, the semicircle on top, and additional balloons that can fit in the corners of the semicircle.

Figure 25: An example of a programming problem with an understanding error.

# G    Consideration for Social Impact

Certainly, it is essential to point out that as AI performs increasingly well on our benchmark, potentially even surpassing human capabilities, there are some potential ethical and moral risks that require collective oversight.

# H    Limitations and Future Work

Despite the value of this benchmark, there remains work to be done in the future. Firstly, our benchmark inevitably introduces some noisy problems, we will actively utilize community feedback to continuously refine it. Additionally, we aim to release new versions of the benchmark annually to mitigate issues related to data leakage. Moreover, this benchmark is currently limited to evaluating models' abilities to solve complex problems. In the future, we aspire for AI to assist with complex tasks and demonstrate value in real-world applications such as AI4Science and AI4Engineering rather than just problem-solving. This will be the goal of our future benchmark designs for evaluating AI capabilities. Nonetheless, at present, OlympicArena plays an essential role as a catalyst for further advancements.

