# OpenReview forum: "OlympicArena: Benchmarking Multi-discipline Cognitive Reasoning for Superintelligent AI"
_NeurIPS.cc/2024/Datasets_and_Benchmarks_Track — NeurIPS 2024 Track Datasets and Benchmarks Poster_

### Official Review · Reviewer_6XA5 · 2024-07-14
**Testing advanced capabilities of SOTA models with OlympicArena**

**Rating:** 6
**Confidence:** 5
**Clarity:** The text is well readable and work is…

**Review:**

OlympicArena presents an important contribution in common community effort to test various capabilities of the models. Authors make sure to reach numbers of test tasks comparable to benchmarks like SciEval. They attempt to extend beyond SOTA by proposing ways to evaluate responses of the models step by step, to check the quality of the reasoning, and also proposing scoring along various defined abilities (eg deductive, inductive reasoning, etc, see Fig 3). They also attempt to evaluate level of potential test set contamination by measuring N-gram prediction accuracy. While such benchmarks are valuable extensions, it is not quite clear how much of currently existing issues this work is able to alleviate compared to already available benchmarks like SciEval. Number of test tasks is not substantially larger. Benchmark itself is static, in contrast to some recent works (eg [1]), which does not help to solve problem of test set contamination on longer term after benchmark release.

[1] Mathador-LM: A Dynamic Benchmark for Mathematical Reasoning on Large Language Models
https://arxiv.org/abs/2406.12572

**Strengths:**

As a benchmark, OlympicArena provides another useful set of tasks to measure important model capabilities, with extended scoring along various capability dimensions including types of reasoning, interesting take on evaluating reasoning quality by looking at step by step model output and stresses currently important issues with benchmarks like test set contamination.

**Additional Feedback:**

No additional feedback

**Correctness:**

The work is sound, discussion on limitations and issues on current state of benchmarks in general (see above) should be extended.

**Documentation:**

Benchmark is publicly available in the github repository, including documentation https://gair-nlp.github.io/OlympicArena/

**Ethics:**

No ethical issues

**Limitations:**

See above - the discrepancy between low scores obtained on very simple problems like AIW problem [1] and scores on OlympicArena (for models like GPT4-o) hint that such benchmarks cannot properly spot weaknesses in basic reasoning shown by SOTA models and thus their usefulness for measuring model capabilities and model comparison is questionable.

[1] Alice in Wonderland: Simple Tasks Showing Complete Reasoning Breakdown in State-Of-the-Art Large Language Models, https://arxiv.org/abs/2406.02061

**Opportunities For Improvement:**

In my view, there is a fundamental limitation that concerns claim of this work to be able to properly test capabilities of SOTA models to perform reasoning and also perform model comparison by looking at the obtained scores. Recent works like the proposed AIW problem [1] show clearly that reasoning breakdown happens already on very simple tasks. For instance, also models that claim strong performance like GPT-4o obtain only about 60% correct response rate on very simple AIW problem, while collapsing below 10% on harder AIW+ that is still arguably well below any Olympic levels. At the same time, benchmarks like presented OlympicArena measure about 20%-30% for GPT-4o on tasks that suppose to measure much harder Olympic level cognition. Obviously, there is a clear problem here: any benchmark that claims proper measurement of model capabilities should be able to react sensitively to failures in very basic reasoning and show scores that reveal weaknesses of the tested models that they show on such simple tasks as AIW problem. For instance, 38% measured on OlympicArena for GPT-4o suggests some Olympic level capability, suggesting rather solid reasoning - which is entirely counteracted by only 60% on very simple AIW tasks (that are arguably average elementary school level problems), and strong collapse on AIW+ going close to 0 correct response rates (that is arguably still well below Olympic levels, and certainly well below difficulty level of computer science olympic tasks where GPT-4o performance is also reported to be low in the author's work), suggesting rather weak reasoning. This discrepancy makes it rather questionable whether scores obtained in such benchmarks like OlympicArena can be used to argue for reflection of model capabilities or their proper comparison - this is though valid for any similar work, not just to specific OlympicArena effort. Authors still should discuss this fundamental matter of Olympic level benchmarks not being able to catch weaknesses in basic reasoning of SOTA models.

Another limitation is as mentioned above the static nature of the benchmarks. Some recent works (eg [2]) aim to solve problem of test set contamination systematically by introducing dynamic benchmarking. Authors should discuss how their effort may line up into that important direction.

[1] Alice in Wonderland: Simple Tasks Showing Complete Reasoning Breakdown in State-Of-the-Art Large Language Models, https://arxiv.org/abs/2406.02061

[2] Mathador-LM: A Dynamic Benchmark for Mathematical Reasoning on Large Language Models
https://arxiv.org/abs/2406.12572

**Relation To Prior Work:**

Relation to prior work is properly reflected. As pointed above, discussion on limitations and issues on current state of benchmarks in general should be extended.

**Summary And Contributions:**

In their work, authors present a set of various tasks from different subjects encompassing wide range of disciplines spanning seven fields and 62 international Olympic competitions, aiming at posing a challenging benchmark - OlympicArena - to test advanced reasoning capabilities of SOTA models using Olympiad level problems. Authors include physics, chemistry, biology, geography, astronomy, and computer science as disciplines selecting Olympic level tasks from 62 different Olympic-level competitions. The benchmark comprises in total 11163 problems (eg compared to 15901 of SciEval), categorized into 13 types of answers expression, interval, etc). Authors incorporate so called process-level evaluations that aim to evaluate step-by-step reasoning processes of the tested models. Authors evaluate a number of SOTA models using OlympicArena, reporting that the strongest performers, eg GPT-4o, reach about 40% in overall accuracy. Authors also take special care to check data leakage and test set contamination, providing some insight whether their test tasks were already known to tested models from the pretraining phase.

---

> ### Author Rebuttal · Authors · 2024-08-15
>
> > Another limitation is as mentioned above the static nature of the benchmarks. Some recent works (eg [2]) aim to solve problem of test set contamination systematically by introducing dynamic benchmarking. Authors should discuss how their effort may line up into that important direction.
>
> We appreciate your suggestion regarding dynamic benchmarking to systematically address the issue of test set contamination. While it is true that our benchmark does not dynamically generate problems like Mathador-LM, we have implemented several strategies to mitigate data leakage effectively.
>
> **Contamination:**
>
> Firstly, we have employed N-gram analysis to check for data leakage and found that our dataset exhibits a very low level of contamination. Furthermore, even in instances where potential data leakage was detected, the models were generally unable to correctly solve the problems, indicating that the leakage does not significantly impact model performance (See 4.5). This finding reassures us that our dataset, while static in some respects, maintains a level of integrity in its current form sufficient to provide reliable evaluations.
>
> **Dynamic Benchmarking:**
>
> Regarding dynamic benchmarking, we leverage the real-world nature of the scientific competitions included in our dataset, which are updated annually. This not only maintains the relevance and real-world applicability of our benchmark but also allows us to continuously introduce new challenges. As detailed in Appendix H, we are curating new versions of the benchmark each year to reflect these updates. This process, akin to a form of dynamic updating, helps in keeping our dataset both relevant and challenging. Additionally, we are exploring data synthesis methods to generate new problems, although this approach sometimes sacrifices some diversity and real-world accuracy.
> In conclusion, while our benchmark does not currently employ dynamic problem generation like Mathador-LM, these steps ensure that it remains updated and effective at measuring the evolving capabilities of AI systems in a realistic and scientifically rigorous manner.

---

> ### Author Rebuttal · Authors · 2024-08-15
>
> Thank you for the effort in reviewing our work and pointing out such constructive comments.
>
> > In my view, there is a fundamental limitation that concerns claim of this work to be able to properly test capabilities of SOTA models to perform reasoning and also perform model comparison by looking at the obtained scores. Recent works like the proposed AIW problem [1] show clearly that reasoning breakdown happens already on very simple tasks. For instance, also models that claim strong performance like GPT-4o obtain only about 60% correct response rate on very simple AIW problem, while collapsing below 10% on harder AIW+ that is still arguably well below any Olympic levels. At the same time, benchmarks like presented OlympicArena measure about 20%-30% for GPT-4o on tasks that suppose to measure much harder Olympic level cognition. Obviously, there is a clear problem here: any benchmark that claims proper measurement of model capabilities should be able to react sensitively to failures in very basic reasoning and show scores that reveal weaknesses of the tested models that they show on such simple tasks as AIW problem. For instance, 38% measured on OlympicArena for GPT-4o suggests some Olympic level capability, suggesting rather solid reasoning - which is entirely counteracted by only 60% on very simple AIW tasks (that are arguably average elementary school level problems), and strong collapse on AIW+ going close to 0 correct response rates (that is arguably still well below Olympic levels, and certainly well below difficulty level of computer science olympic tasks where GPT-4o performance is also reported to be low in the author's work), suggesting rather weak reasoning. This discrepancy makes it rather questionable whether scores obtained in such benchmarks like OlympicArena can be used to argue for reflection of model capabilities or their proper comparison - this is though valid for any similar work, not just to specific OlympicArena effort. Authors still should discuss this fundamental matter of Olympic level benchmarks not being able to catch weaknesses in basic reasoning of SOTA models.
>
> Thank you for your insightful comment regarding the ability of state-of-the-art (SOTA) models to perform reasoning and the implications for model comparison in benchmarks like OlympicArena. We acknowledge that even as models achieve breakthroughs on complex mathematical or Olympiad-level tasks, they may still struggle with very simple tasks, such as commonsense reasoning or basic numerical comparisons like determining whether 9.11 is greater than 9.9. This discrepancy indeed poses significant questions about the overall reasoning capabilities of these models. However, our benchmark specifically focuses on solving complex scientific problems, akin to those in benchmarks like MATH[1], GSM-8K[2], GPQA[3], and SciBench[4], which are designed to evaluate scientific problem-solving. While it is true that SOTA models exhibit weaknesses in simpler tasks—a reality that other benchmarks are set up to assess—OlympicArena aims to probe the depth of reasoning models can apply in the context of sophisticated scientific questions. The value of our benchmark lies in its ability to evaluate whether models can perform complex reasoning required for scientific exploration with multimodal input at an Olympiad level. Advancements in performance on such benchmarks can foster improvements in how models undertake independent and advanced scientific inquiries. Furthermore, our benchmark has a direct and beneficial relationship with AI4Science initiatives, as it provides a rigorous framework to test and enhance the capabilities of AI systems in conducting autonomous scientific research.
>
> We recognize the importance of the very simple AIW tasks, which indeed represent a crucial area where SOTA models need improvement. However, the ability to tackle these elementary problems and the challenges presented in OlympicArena are not mutually exclusive. Each set of tasks provides unique insights into the models’ capabilities. We will discuss this aspect more explicitly in our paper to clarify that while OlympicArena tests high-level reasoning, it is equally important for models to handle basic reasoning tasks effectively.
>
> [1] Measuring Mathematical Problem Solving With the MATH Dataset
>
> [2] Training Verifiers to Solve Math Word Problems
>
> [3] GPQA: A Graduate-Level Google-Proof Q&A Benchmark
>
> [4] SciBench: Evaluating College-Level Scientific Problem-Solving Abilities of Large Language Models

---

### Official Review · Reviewer_eKD6 · 2024-07-16
**Benchmark for cognitive reasoning and problem solving in LLMs and LMMs**

**Rating:** 7
**Confidence:** 3

**Review:**

The paper introduces a new benchmark and data set (for the benchmark) for evaluating problem solving and reasoning abilities of LLMs/LMMs in a wide range of tasks. The benchmark is, to my knowledge and understanding, an improvement to the current SoTA. The paper is also well written and presents the content in a relatively clear way. The paper also establishes a baseline that would be useful for assessing problem-solving and reasoning abilities for future LLM/LMMs.

Considering that problem solving and reasoning in current LLM/LMMs are relatively weak, I think this benchmark is a significant contribution to future research and improvements to such models.

**Strengths:**

The paper introduces a new benchmark for evaluating reasoning abilities of LLMs/LMMs in a variety of different tasks. The benchmark is an improvement, compared to the current SotA, in terms of the amount of problems available, difficulty, data leakage, and more. The paper also presents a comprehensive analysis of current SoTA models' performance on the benchmark, providing insight into the limitations in problem solving and reasoning abilities of current LLM/LMMs.

Given the recent advances and popularity, the paper is highly relevant to anyone interested in LLMs/LMMs. Thus, I think most of the NeurIPS community would find this paper interesting.

**Additional Feedback:**

I would consider summarize some of the more important details that are currently in the appendix, in the main paper.

**Clarity:**

The paper is in overall well written. I also think that the authors have presented and visualized their results and findings in a very nice way.

However, I have a few minor remarks: I found the left part of figure 1 confusing. I think it would be useful to provide a more detailed explanation of the figure in the caption. Also, some of the text in Figure 1 and Figure 3 is small and may be unintelligable on printed paper (A4).

**Correctness:**

I think it is thoughtful that the authors have included data leakage detection to evaluate the objectiveness and applicability of the benchmark to a certain model.

Ideally, I think there should be some error numbers in Table 3 (could be written as +-).

**Documentation:**

The dataset and benchmark are documented sufficiently, with a data card describing data collection, maintenance and organization. The dataset for the benchmark is available through Huggingface, while the benchmark is in a Github repository. Both sites contain clear and understandable instructions for installation and setup.

**Ethics:**

To my understanding, the dataset that is used for the benchmark does not contain any information subjected to privacy, copyrights or information that can be harmful. I don't see any other immediate ethical concerns with this submission either.

As a remark: considering that this benchmark could potentially be used to make LLM/LMMs better at problem solving, and perhaps even more intelligent, it may be relevant to discuss whether supporting future research into this topic is the right thing to do or not (I think that could be a nice debate for the upcoming conference).

**Limitations:**

The authors have included many different models. However, I would have liked to see evaluations of Llama 3 as well.

**Opportunities For Improvement:**

I think the paper in general makes too many references to the appendix, also on some details that should preferably be mentioned in the main paper. For example, where and how are the various olympic competitions (in the dataset) being held? I understand that this is due to space restrictions.

I think the difficulty vs. complexity aspects regarding the problems in the benchmark could have been mentioned, as many problems can be difficult to solve due to their complexity (vs. size scaling). Such as certain matrix operations (math), NP-class decision problems (CS), etc.

**Relation To Prior Work:**

The relation to prior work seems well described, with references to many existing related benchmarks. The authors clearly describe the difference between their own and earlier contributions from others.

There is a recent benchmark published at CVPR 2024 called SEED-Bench that are not mentioned in the paper. However, this is likely because CVPR was held in June, after the NeurIPS 2024 submission deadline. I think the differences between OlympicArena and SEED-Bench should be discussed explicitly in the paper when the authors have the chance to update it.

**Summary And Contributions:**

The paper introduces a new benchmark for LLMs and/or LMMs called OlympicArena, which includes a large number of difficult problems, covering a wide range of areas and 62 international olympic competitions. The benchmark is larger and more extensive than the ones that are currently available, and addresses many of the areas where current SoTA LLM/LMMs are not performing very well. The paper also contain an analysis of SoTA models that are tested on the benchmark, providing insights into the models' shortcomings and establishing baselines for future research.

---

> ### Author Rebuttal · Authors · 2024-08-15
>
> Thanks for your time and encouraging review! We respond to your comments below.
>
> > I think the paper in general makes too many references to the appendix, also on some details that should preferably be mentioned in the main paper. For example, where and how are the various olympic competitions (in the dataset) being held? I understand that this is due to space restrictions.
>
> We are grateful for the reviewer’s understanding regarding the extensive references to the appendix in our paper. Due to the space restrictions imposed by the submission guidelines, we had to transfer several sections to the appendix, which unfortunately led to the compression of some critical information in the main text. In future versions, where we have more space available, we will ensure that key information (e.g., “where and how the competitions are held”).
>
> > I think the difficulty vs. complexity aspects regarding the problems in the benchmark could have been mentioned, as many problems can be difficult to solve due to their complexity (vs. size scaling). Such as certain matrix operations (math), NP-class decision problems (CS), etc.
>
> Thank you for your insightful comment . We agree that this distinction is crucial, as complexity can often make a problem inherently difficult, regardless of its size or apparent scope. Indeed, we have classified the difficulty of each problem into three distinct levels: knowledge recall, concept application, and cognitive reasoning, as detailed in Line 173 and Table 1 of our paper. These levels are designed to reflect not only the depth of knowledge required but also the complexity of the cognitive processes involved in solving the problems. This classification helps in understanding the multifaceted challenges posed by different tasks within the benchmark, from simple recall to complex reasoning and decision-making in fields like mathematics and computer science, where problems can involve intricate matrix operations or NP-class decision issues. In subsequent versions of our paper, we will explicitly mention and analyze these concepts to provide a clearer and more detailed understanding of how difficulty vs. complexity influence problem-solving in the benchmark. Thank you for highlighting this aspect, which will certainly enrich our discussion and analysis in future revisions.
>
>
> > The authors have included many different models. However, I would have liked to see evaluations of Llama 3 as well.
>
> Our benchmark is an ongoing project that we continuously maintain and update. We have not only tested the model you mentioned (LLama3), but also evaluated several other newly released models, such as Claude-3.5-Sonnet, Qwen2-72B-Instruct, Gemini-1.5-Pro, etc. The results have been added to our project website (https://gair-nlp.github.io/OlympicArena/). We will continue to update the benchmark with the performance of more models as they become available.
>
> > Ideally, I think there should be some error numbers in Table 3 (could be written as +-).
>
> We appreciate your suggestion to include error numbers to represent variability in our experimental results, as this would indeed enhance the rigor and reliability of the presented data. However, due to the significant costs associated with API calls and computational resources, conducting multiple runs across a benchmark of 10k+ data points for each model is financially and logistically challenging. We plan to explore more cost-effective methods or funding opportunities in future work to include error margins and ensure a more robust analysis.
>
> > I found the left part of figure 1 confusing. I think it would be useful to provide a more detailed explanation of the figure in the caption. Also, some of the text in Figure 1 and Figure 3 is small and may be unintelligable on printed paper (A4).
>
> Thank you for kindful suggestion. In the next version of the paper, we will enhance the captions for the figures to provide a more detailed explanation, which should clarify their content. Additionally, we will adjust the font sizes to ensure that all text is legible and easily understandable in printed form.
>
> > There is a recent benchmark published at CVPR 2024 called SEED-Bench that are not mentioned in the paper. However, this is likely because CVPR was held in June, after the NeurIPS 2024 submission deadline. I think the differences between OlympicArena and SEED-Bench should be discussed explicitly in the paper when the authors have the chance to update it.
>
> Thank you for bringing SEED-Bench to our attention. We also appreciate your understanding that the CVPR conference, where SEED-Bench was introduced, took place after the NeurIPS submission deadline. SEED-Bench represents a significant and comprehensive multimodal benchmark that evaluates LMMs on various visual scenarios, such as scene understanding and action recognition, using multiple-choice questions. In contrast, our OlympicArena benchmark is specifically designed to assess complex scientific problem-solving and reasoning capabilities, demanding a higher degree of specificity and depth in reasoning skills. We will include a discussion of SEED-Bench as related work in the next version of our paper, highlighting these differences and the unique focus of OlympicArena.

---

### Official Review · Reviewer_vFpF · 2024-08-05

**Rating:** 7
**Confidence:** 3
**Correctness:** Yes
**Clarity:** Yes

**Review:**

Originality: The paper presents a novel and original contribution by introducing the first Olympic-level, multidisciplinary benchmark that comprehensively evaluates the cognitive reasoning abilities of LLMs and LMMs. The combination of challenging problems, multimodal and bilingual support, and fine-grained evaluation sets it apart from existing benchmarks.

Significance: The OlympicArena benchmark has the potential to drive significant advancements in the development of more intelligent and capable AI, with implications for various fields, including science, engineering, and beyond.

Clarity: The paper is well-written and clearly organized, with a clear presentation of the benchmark's design, data collection and annotation processes, experimental setup, and evaluation results. The authors provide sufficient details and explanations to enable reproducibility and facilitate further research.

Quality: The overall quality of the work is high. The authors have taken great care in curating the benchmark, ensuring data quality through a multi-step validation process and addressing potential data leakage issues. The experimental design is sound, and the results are thoroughly analyzed and discussed.

**Strengths:**

1. Comprehensive and Challenging: The benchmark covers a wide range of scientific disciplines and includes a large number of problems, making it a comprehensive and challenging test for AI models. The inclusion of Olympic-level problems ensures a high level of difficulty, pushing the boundaries of current AI capabilities.

2. Multimodal and Bilingual: The benchmark supports both text-only and interleaved text-image modalities, catering to the diverse capabilities of modern AI systems. The bilingual nature of the benchmark (English and Chinese) further enhances its accessibility and applicability.

3. Fine-grained Evaluation: The evaluation mechanism goes beyond simple answer-based assessment, incorporating process-level evaluation to scrutinize the intermediate reasoning steps. This provides a deeper understanding of the models' cognitive processes and helps identify potential weaknesses in their reasoning abilities.

**Additional Feedback:**

NA

**Documentation:**

Yes

**Limitations:**

Yes

**Opportunities For Improvement:**

1. Limited Scope: The current benchmark focuses primarily on problem-solving abilities. While this is an important aspect of cognitive reasoning, future versions could expand the scope to include other cognitive tasks, such as memory recalling, creative problem-solving, scientific discovery, and real-world applications.

2. Lack of Evaluation on open-sourced models such as LLama3 models, Gemma 2 models etc. It would be great to see how these mainstream models perform on this dataset.

3. Self-consistency is generally useful on mathematical reasoning. Have authors tried any SC experiments on these datasets?

**Relation To Prior Work:**

Yes

**Summary And Contributions:**

The paper introduces OlympicArena, a new benchmark designed to evaluate the cognitive reasoning capabilities of LLMs and Large Multimodal Models. The benchmark comprises a vast collection of 11,163 problems sourced from 62 international Olympic competitions across seven scientific disciplines, including mathematics, physics, chemistry, biology, geography, astronomy, and computer science. The problems are available in both English and Chinese and include both text-only and interleaved text-image modalities. The benchmark also includes a fine-grained evaluation mechanism that assesses not only the accuracy of the final answers but also the correctness of the intermediate reasoning steps, providing a more comprehensive understanding of the models' cognitive processes. The authors conducted extensive experiments on various state-of-the-art LMMs and LLMs, and the results reveal that even the most advanced models struggle with the complex, multidisciplinary problems in the benchmark, highlighting the need for further research to improve the cognitive reasoning abilities of AI systems.

---

> ### Author Rebuttal · Authors · 2024-08-15
>
> We sincerely appreciate the insightful comments and suggestions from you!
>
> > Limited Scope: The current benchmark focuses primarily on problem-solving abilities. While this is an important aspect of cognitive reasoning, future versions could expand the scope to include other cognitive tasks, such as memory recalling, creative problem-solving, scientific discovery, and real-world applications.
>
> This is a good point! We acknowledge that the scope of our benchmark primarily focuses on complex, multidisciplinary problem-solving tasks that demand advanced reasoning capabilities. However, we believe that creating separate benchmarks tailored to specific tasks requiring different cognitive skills is a more effective approach. For instance, benchmarks like AgentBench[1], GAIA[2], and VisualWebArena[3] excellently assess the planning capabilities of LLM agents and their ability to use tools for real-world tasks. Similarly, Swe-Bench[4] evaluates debugging capabilities in real-world software development, and benchmarks like ChemLLMBench[5] evaluate real-world scientific discovery tasks. Furthermore, we contend that successfully addressing the complex reasoning challenges in our benchmark can significantly contribute to AI’s ability to tackle more intricate real-world tasks. We consider the application of our benchmark to more real-world scientific scenarios, such as scientific discovery, an important direction for future work.
>
> [1] AgentBench: Evaluating LLMs as Agents
>
> [2] GAIA: A Benchmark for General AI Assistants
>
> [3] VisualWebArena: Evaluating Multimodal Agents on Realistic Visual Web Tasks
>
> [4] SWE-Bench: Can Language Models Resolve Real-world Github Issues?
>
> [5] What can Large Language Models do in chemistry? A comprehensive benchmark on eight tasks
>
>
>
> > Lack of Evaluation on open-sourced models such as LLama3 models, Gemma 2 models etc. It would be great to see how these mainstream models perform on this dataset.
>
> We appreciate your suggestion to include evaluations of open-sourced models such as LLama3 and Gemma 2 in our benchmark. Initially, we chose not to include LLama3 in the paper’s baseline evaluations due to its limited support for non-English languages, which is a significant component of our bilingual benchmark. Additionally, Gemma 2 was released after the NeurIPS submission deadline.
>
> However, our benchmark is an ongoing project that we continuously maintain and update. We have not only tested the models you mentioned (LLama3, Gemma2), but also evaluated several other newly released models, such as Claude-3.5-Sonnet, Qwen2-72B-Instruct, Gemini-1.5-Pro, etc. The results have been added to our project website (https://gair-nlp.github.io/OlympicArena/). We will continue to update the benchmark with the performance of more models as they become available.
>
>
> > Self-consistency is generally useful on mathematical reasoning. Have authors tried any SC experiments on these datasets?
>
> Thank you for raising the question about the role of self-consistency in mathematical reasoning within our dataset. Self-consistency is indeed a valuable technique in reasoning tasks and can significantly enhance model performance. Due to the constraints of time and budget at the time of submission, we were unable to conduct extensive self-consistency experiments across our entire dataset. However, we did explore the impact of self-consistency on the performance of the current state-of-the-art closed-source model, GPT-4o, within our validation set (CS subject already employs an unbiased pass@k metric with built-in multiple sampling, so we excluded it in the following table).
>
> | **Metric**                | **n=1** | **n=3** | **n=5** |
> |---------------------------|---------|---------|---------|
> | **Total correct rate**     | 32.29%  | 34.80%  | 34.95%  |
> | **Subject correct rate**   |         |         |         |
> | Math                       | 33.20%  | 35.25%  | 35.66%  |
> | Physics                    | 32.22%  | 34.44%  | 34.44%  |
> | Chemistry                  | 24.62%  | 32.31%  | 27.69%  |
> | Biology                    | 31.75%  | 33.33%  | 34.92%  |
> | Geography                  | 42.65%  | 42.65%  | 45.59%  |
> | Astronomy                  | 31.11%  | 34.44%  | 34.44%  |
>
> As these results indicate, the application of self-consistency does lead to a noticeable improvement in performance, which is consistent with findings in other related research. Although we conducted these experiments with a relatively small n due to API costs and computational resource limitations, we believe that increasing the sampling size could yield further improvements in accuracy, though the benefits would likely plateau. We sincerely thank the reviewer for their valuable suggestion. We commit to continuing this line of investigation in future work, ensuring a more comprehensive evaluation across our entire dataset.

---

### Decision · Program_Chairs · 2024-09-26

**Decision:**

Accept (Poster)

**Comment:**

This paper presents a dataset of 11,163 problems for assessing the cognitive reasoning abilities of foundation models.  The datasets is well documented and large.  The reasoning capabilities of LLMs needs to be evaluated rigorously and this dataset could be a useful tool for other researchers.  The reviewers agree that this is a strong submission and should be accepted to the conference.  I would encourage the authors to integrate any additional limitations that they forsee with this dataset into their paper.  Every datasets has limitations and the creators tend to know these best.  Some of the plots are a little hard to read (small fonts) these could be improved.